# Colorectal Cancer Bioengineered Microtissues as a Model to Replicate Tumor-ECM Crosstalk and Assess Drug Delivery Systems In Vitro

**DOI:** 10.3390/ijms24065678

**Published:** 2023-03-16

**Authors:** Alessia La Rocca, Vincenza De Gregorio, Elena Lagreca, Raffaele Vecchione, Paolo Antonio Netti, Giorgia Imparato

**Affiliations:** 1Center for Advanced Biomaterials for Health Care (CABHC), Istituto Italiano di Tecnologia, 80125 Napoli, Italy; larocca.alessia@hsr.it (A.L.R.); elena.lagreca@iit.it (E.L.); raffaele.vecchione@iit.it (R.V.); paolo.netti@iit.it (P.A.N.); 2San Raffaele Diabetes Research Institute, IRCCS San Raffaele Scientific Institute, 20132 Milan, Italy; 3Interdisciplinary Research Centre on Biomaterials (CRIB), University of Naples Federico II, 80125 Naples, Italy; vincenza.degregorio@unina.it; 4Department of Biology, University of Naples Federico II, Complesso Universitario Monte S. Angelo, 80126 Naples, Italy; 5Department of Chemical Materials and Industrial Production (DICMaPI), University of Naples Federico II, 80125 Naples, Italy

**Keywords:** tumor microenvironment (TME), extracellular matrix (ECM), cancer associated fibroblasts (CAF), curcumin-loaded nanoemulsion (CT-NE-Curc), 5-Fluorouracil (5-FU)

## Abstract

Current 3D cancer models (in vitro) fail to reproduce complex cancer cell extracellular matrices (ECMs) and the interrelationships occurring (in vivo) in the tumor microenvironment (TME). Herein, we propose 3D in vitro colorectal cancer microtissues (3D CRC μTs), which reproduce the TME more faithfully in vitro. Normal human fibroblasts were seeded onto porous biodegradable gelatin microbeads (GPMs) and were continuously induced to synthesize and assemble their own ECMs (3D Stroma μTs) in a spinner flask bioreactor. Then, human colon cancer cells were dynamically seeded onto the 3D Stroma μTs to achieve the 3D CRC μTs. Morphological characterization of the 3D CRC μTs was performed to assess the presence of different complex macromolecular components that feature in vivo in the ECM. The results showed the 3D CRC μTs recapitulated the TME in terms of ECM remodeling, cell growth, and the activation of normal fibroblasts toward an activated phenotype. Then, the microtissues were assessed as a drug screening platform by evaluating the effect of 5-Fluorouracil (5-FU), curcumin-loaded nanoemulsions (CT-NE-Curc), and the combination of the two. When taken together, the results showed that our microtissues are promising in that they can help clarify complex cancer–ECM interactions and evaluate the efficacy of therapies. Moreover, they may be combined with tissue-on-chip technologies aimed at addressing further studies in cancer progression and drug discovery.

## 1. Introduction

In the last decades, enormous progress in experimental and molecular cancer research, new anticancer drug development, and drug testing have been achieved, but cancer continues to be a leading disease in mortality worldwide. Colorectal cancer (CRC) is a malignant disease that develops from the mucosa of the colon and rectum and is one of the most common cancers worldwide [1], ranking third in terms of incidence (10.2% of all cancer cases worldwide), and is the second most common cause of cancer mortality, immediately behind lung, liver, and stomach cancer (9.2% of all cancer mortality) [2]. More evidence suggests that the evolution from a healthy colon to an invasive colon carcinoma is supported by the local tumor microenvironment (TME), which plays an essential role in cancer growth and propagation [3]. The TME is characterized by crucial bidirectional crosstalk between cancer cells and the surrounding stromal tissues in which the extracellular matrix (ECM) orchestrates the signaling among different cell types (fibroblasts, endothelial cells, and immune and inflammatory cells) [4]. More specifically, paracrine signaling, including the release of cytokines and proteolytic enzymes, alters ECM remodeling, and the transformation of healthy cells within reactive and tumor stromal cells, such as fibroblasts, endothelial cells, and mesenchymal stem cells, promoting cancer growth, local invasion, and distant metastasis formation [5]. Thus, it is imperative to understand the mechanisms underlying the dynamic interactions between cancer cells and the other cells of the TME to develop an in vitro tumor model that can be used to investigate cancer progression and test new drugs and therapies against CRC.

Even if clinically relevant to better reproducing the TME and maintaining a high degree of genomic complexity, human resected tumors that are implanted in immunocompromised mice, named patient-derived xenograft (PDX) models, present a low grafting rate, and the site and frequency of PDX CRC may vary from that seen in the patient [6]. In addition, PDX, as with other in vivo models, suffers from the high cost of animal models, ethical issues, and the lack of predictability for human responses to drugs and disease [7]. On the other hand, in vitro 2D cell culture models are not able to replicate the 3D architecture and complexity of ECM and fail to replicate cell heterogeneity and the interactions occurring among cells and between the cell and the ECM in the TME [8].

In recent years, in vitro 3D tumor models have been developed to closely mimic the complexity of the TME in CRC. Multicellular tumor spheroids and organoids constitute invaluable tools for recapitulating the most important mechanisms underlying the complex interactions between cells and the ECM in the TME, including altered ECM remodeling and stromal cell activation. In this respect, previous studies have displayed an increase in collagen degradation and the deposition in collagen ECM hydrogel-containing stromal cells in response to the factors secreted by colon cancer cells [9], and others have observed that normal colonic fibroblasts (NCFs) are converted into cancer-associated fibroblasts (CAFs) when cocultured in spheroids with colon cancer cells [10]. Furthermore, in vitro 3D tumor models have been used to screen drugs and drug delivery systems (DDSs), improving the identification of potential therapeutic targets in the TME, with the aim of blocking or suppressing the ECM-mediated mechanisms involved in tumor progression [11]. However, most of the existing 3D models often do not correctly recapitulate TME physiology, complexity, and the mechanisms underlying the interactions that occur in the native tissue, owing to the use of the exogenous matrix as surrogate for the native ECM. It has been proved that when fibroblasts are induced to produce and assemble their own ECM using the porous microbeads-based approach, they succeed in forming 3D tumor models that better recapitulate the 3D structure and organization of the native tumor tissue [12].

5-Fluorouracil (5-FU) is widely used as a chemotherapeutic agent for the treatment of metastatic colorectal cancer. It acts by blocking cancer cell proliferation and induces apoptosis via the incorporation of its metabolites into DNA and RNA as a thymidylate synthase inhibitor to block dTMP synthesis [13]. Even though 5-FU is widely used to treat colorectal cancer, 5-FU is associated with several adverse risks, such as cardiotoxicity, owing to the high dosage needed because of the progressive resistance of cancer cells. Nutraceuticals are promising candidates to modulate the TME and, thus, support chemotherapy in CRC, preventing the adverse risks on different organs. Among these, curcumin has exhibited potential preventive and therapeutic effects, such as anti-inflammatory, antioxidant, and, especially, anticancer [14]. It has been shown that curcumin enhances the cytotoxic effects of several chemotherapeutic agents, such as 5-FU in colon cancer cells [15]. However, it is known that many nutraceuticals, including curcumin, are unstable or poorly water-soluble, affecting their efficacy upon oral administration. Indeed, previous studies have demonstrated that oil in water nanoemulsions that are loaded with curcumin protect it from degradation, improving its bioavailability [16,17,18].

In this study, we produced 3D in vitro colon cancer tumor microtissues (3D CRC μTs) to reproduce a more relevant TME and investigate its role in cancer progression. Our 3D μTs were realized by seeding colon cancer cells (HCT-116 cell line) and/or normal fibroblasts (NFs) into porous biodegradable microbeads in a dynamic and controlled culture system. Our approach allowed us to induce the fibroblasts to synthesize and assemble their own ECM, guiding the cells in synthesizing and assembling a complex 3D matrix and better recapitulating the native TME [12]. In the first instance, we exploited 3D monocultured CRC μTs (3D HCT-116 μTs), 3D co-cultured CRC μTs (3D HCT-116/NFs μTs), and 3D Stroma μTs (NFs alone), and assessed the difference in terms of cell growth, matrix deposition, ECM remodeling, and the bidirectional crosstalk between cancer cells and fibroblasts. Once we established a configuration that better replicated the main features of CRC in terms of cell growth, matrix deposition, ECM remodeling, the bidirectional crosstalk between cancer cells, and the fibroblasts, the 3D CRC μTs were used as potential in vitro models for drug and/or nutraceutical-loaded DDS testing. In this respect, we assessed the 3D CRC μTs as a model for the screening of drugs and DDSs by investigating the effect of 5-FU, curcumin-loaded nanoemulsion (CT-NE-Curc), and their combination. It was observed that the 2D cell cultures were still useful for a deep understanding of the mechanisms behind the related therapeutic efficacy, although our in vitro 3D CRC μTs emerged as a promising platform to investigate the crucial processes of the interplay between cancer cells-fibroblasts and cellular ECM in the TME and the effects of new drugs and/or nutraceuticals on the cancer progression.

## 2. Results

### 2.1. Time Evolution of In Vitro 3D Colorectal Cancer Microtissues

In this study, we developed three different configurations of 3D µTs using a cell-seeded microbead approach and monitored their evolution during the 12 days of culturing the 3D co-culture of colorectal cancer microtissues (3D CRC µTs) (consisting of normal fibroblasts (NFs) and HCT-116 cells), the 3D monoculture colorectal cancer microtissues (consisting of HCT-116 cells alone) (3D HCT-116 µTs), and 3D monoculture stroma microtissues (consisting of NFs alone (3D Stroma µTs)). More specifically, we produced two types of 3D CRC µTs: the first consisted of HCT-116 cells, and NFs seeded together on day 0 of culturing (3D CRC Day0 µTs) and the second consisted of HCT-116 cells seeded on 3D Stroma µTs on day 4 of culturing (3D CRC Day4 µTs) (Figure 1).

The confocal images showed that the proliferation of the HCT-116 cells (GFP green signal) in 3D CRC Day0 µTs was higher (Figure 2A–C) than that of the 3D CRC Day4 µTs (Figure 2D–F) at all culture times, as reported in the graph (Figure 2G). In order to investigate ECM production and degradation, we acquired the second harmonic generation (SHG) signal of the unstained collagen fibers, which are endogenously produced in 3D CRC µTs. More specifically, the 3D CRC Day0 µTs showed a low SHG signal on day 8 (Figure 2B), while a rare SHG signal was detected on day 12 (Figure 2C), indicating a drastic reduction in collagen fibers. On the contrary, the SHG signal in the 3D CRC Day4 µTs increased from day 4 to 12 of culturing (Figure 2D–F). Furthermore, we quantitatively evaluated collagen fiber production (collagen fraction %) and the collagen assembly degree (CAD) from the SHG images for the different configurations of the 3D CRC µTs. Our data showed that the collagen fraction increased linearly over time from day 4 to 12 in the 3D CRC Day4 µTs (Figure 2H) compared to the 3D CRC Day0 µTs, in which the collagen fraction increased from day 4 to 8 and then dramatically decreased until the end of culturing. Further, the 3D CRC Day0 µTs showed high CAD values on day 4 (1.8 ± 0.1) and day 8 (47.5 ± 5.7), and these values sharply decreased on day 12 (15.6 ± 3.6). On the contrary, the 3D CRC Day4 µTs displayed an increased assembly degree of collagen from day 4 (18.1 ± 2.4) to day 8 (32.5 ± 4.6), and this remained constant until day 12 of culturing (34 ± 3.4) (Figure 2I). These results display that the 3D CRC Day0 µTs lack a structured and organized ECM compared to the 3D CRC Day4 µTs.

Moreover, to increase ECM production in the 3D CRC Day4 µTs, we added Ascorbic Acid (AA) at a concentration of 50 µg/mL during the dynamic culturing of the 3D CRC Day4 µTs until day 12 of culturing in order to stimulate the human fibroblasts to synthesize the collagen matrix. Therefore, we compared the 3D CRC Day4 µTs with AA (w AA) (Figure 3A–C) or without AA (w/o AA) (Figure 3D–F) in order to investigate both the collagen fibers production and cancer cell proliferation. Our data showed that there were no relevant differences in cancer cell proliferation from day 4 to day 12 between the two configurations analyzed (Figure 3G), suggesting that AA did not interfere with cancer cell viability. The confocal images highlighted a slight increase in the SHG intensity signal from day 4 to day 8 and a strong signal in the 3D CRC Day4 μTs with AA at day 12 (Figure 3A–C), compared to 3D CRC Day4 μTs without AA (Figure 3D–F). Furthermore, the results of the collagen fraction analysis showed that ECM production slightly increased from day 4 to day 8 in the 3D CRC Day4 µTs without AA, keeping constant until day 12 of culturing (Figure 2H). On the contrary, the 3D CRC Day4 µTs with AA exhibited a very low level of ECM production on day 4 and day 8, but it showed a strong increase in collagen fraction on day 12 (Figure 3H). Of note, the CAD showed the same trend of collagen fraction between the two samples analyzed, with a slight increase in collagen fiber assembly degree in the 3D CRC Day4 µTs without AA from day 4 (18.1 ± 2.4) to day 12 (34 ± 3.4), compared to the strong increase in CAD in the 3D CRC Day4 µTs with AA at day 12 (125.2 ± 11.4) (Figure 3I).

Further analysis was carried out by comparing the 3D CRC Day4 µTs with AA with the 3D HCT-116 µTs and the 3D Stroma µT monocultures (as controls). The confocal images showed a low proliferation rate of HCT-116 cells in the 3D CRC Day4 µTs with AA on days 4 and 8 of culturing (Figure 4A,B), compared to the 3D HCT-116 µTs (Figure 4D,E). However, the cancer cell number in the 3D CRC Day4 µTs with AA (Figure 4C) and 3D HCT-116 µTs (Figure 4F) were very similar on day 12, as reported in Figure 4J. Then, we compared the 3D CRC Day4 µTs with AA with the 3D Stroma µTs in terms of ECM deposition. The confocal images exhibited a low SHG signal for the collagen fibers in the 3D CRC Day4 µTs with AA (Figure 4A–C), compared to the 3D Stroma µTs (Figure 4G–I) on days 4, 8, and 12 of culturing. This aspect was also shown in the collagen fraction analysis, where ECM production increased in the 3D Stroma µTs from day 4 to 12, in contrast to the 3D CRC Day4 µTs with AA (Figure 4K). An analysis of the collagen fraction on the 3D Stroma µTs revealed that the collagen fibers were continuously assembled from day 4 to day 8, compared to the 3D CRC Day4 µTs with AA (Figure 4L). Interestingly, we noticed that the 3D CRC Day4 µTs with AA displayed a slight increase in CAD on day 12 of culturing (125.2 ± 11.4), which is in contrast to the 3D Stroma µTs (104.1 ± 10.4) (Figure 4L). Additionally, a higher magnification of the 3D CRC Day4 µTs and the 3D Stroma µTs was acquired to better show the different structure and organization of the endogenous ECM (Appendix A). On the basis of these results, we selected the 3D CRC Day4 µTs with AA configuration, in which the distribution and proliferation of the cancer cells, collagen fraction, and assembly better recapitulate the in vivo TME of colorectal cancer. In the next steps, in order to evaluate the crosstalk between cancer cells and fibroblasts in our models, we compared the 3D CRC µTs with the 3D Stroma µTs.

### 2.2. Colorectal Cancer Cells Promote Continuous Extracellular Matrix Remodeling in 3D CRC µTs

In order to investigate the ability of cancer cells to remodel the surrounding endogenous ECM, we analyzed the distribution of collagen fiber orientation in the tumor model (3D CRC µTs) and the 3D stroma µT monocultures. The SHG images acquired after 12 days of culturing showed the presence of aligned collagen fibers in some regions of the 3D CRC μTs when compared to the 3D Stroma μTs (Figure 5A,B and insets, respectively). Moreover, we evaluated the deposition and distribution of collagen fibers in the different regions of both samples by using the color map of orientations, showing a preferential orientation for the collagen fibers, stained green-yellow, at −30°/−60° in the 3D CRC μTs (Figure 5C, red perimeter). Instead, the 3D Stroma µTs exhibited randomly distributed collagen fibers with no preferential orientation (Figure 5D, blue perimeter). In order to quantify the alignment of the collagen fibers in the 3D CRC µTs or 3D Stroma µTs, we measured the coherency index. Interestingly, we highlighted high-grade collagen fiber orientation in the 3D CRC µTs (0.20 ± 0.013) when compared to the 3D Stroma µTs (0.12 ± 0.002) (Figure 5E). Additionally, we observed that the 3D CRC µTs exhibited a preferential range of collagen fiber orientation of about −20°/−35° (Figure 5F). This range was also found within some of the HCT-116 cell clusters that were aligned along the collagen fibers tracks (Figure 5G).

Further, the histological cross-section stained with Hematoxylin/Eosin (H/E) revealed that the 3D CRC μTs exhibited significantly higher ECM degradation (Figure 6A,C) compared to the 3D Stroma μTs, in which ECM maintained its integrity and structure (Figure 6B,D). Moreover, we observed that while some cancer cells had a round shape, when resembling epithelial cells on the top layer of the 3D CRC μTs (Figure 6C, white arrows), the others showed a spindle shape, resembling fibroblasts, when they were embedded in the residual ECM (Figure 6C, black arrows). In the meantime, we evaluated whether matrix metalloproteinases (MMPs) secretion, the enzymes that degrade ECM and non-ECM molecules, was associated with the continuous ECM remodeling in our tumor model. In this regard, we examined the Matrix metalloproteinase-9 (MMP-9) expression in the 3D CRC μTs (Figure 6E,G,I) and the 3D Stroma μTs (Figure 6F,H,J) by qualitative immunostaining. The results showed a strong MMP-9 signal in the 3D CRC μTs (Figure 6G) compared to a weak signal in the 3D Stroma μTs (Figure 6H). Indeed, we observed that the MMP-9 signal was localized within the cytoplasm in both the fibroblasts and the HCT-116 cells in the 3D CRC μTs, especially in the cell protrusions of the plasma membrane (Figure 5G: yellow and white arrows for HCT-116 cells and fibroblasts, respectively).

### 2.3. Fibroblasts Reprogramming In Vitro: Activation of Normal Fibroblasts into Cancer-Associated Fibroblasts in 3D CRC μTs

In order to determine whether the fibroblasts in the 3D CRC µTs underwent the necessary change to become Cancer-associated Fibroblasts (CAFs), we estimated the expression of α-smooth muscle actin (αSMA) and Fibroblast Activation Protein α (FAPα), two specific markers used for CAF identification. Immunofluorescence staining was carried out to highlight the αSMA and FAPα signals and their localization in the 3D CRC µTs (Figure 7A–C and Figure 7G–I, respectively) and the 3D Stroma µTs (Figure 7D–F and Figure 7J–L, respectively). The confocal images showed that the αSMA signal was distributed at the cytoplasmic level throughout the fibroblasts, and its signal was significantly higher in the 3D CRC μTs (Figure 7B,C, white arrows) than in the 3D Stroma μTs, where the signal was almost absent (Figure 7E,F). Moreover, we found a high αSMA signal in the HCT-116 cells (Figure 7C, yellow arrows). Additionally, we found a strong signal of FAPα in the 3D CRC μTs (Figure 7H,I) when compared to the 3D Stroma μTs (Figure 7K,L). We revealed a greater and more widespread signal for FAPα in both the fibroblasts and HCT-116 cells in the 3D CRC μTs (Figure 7H,I, white arrows for fibroblasts; yellow arrows for HCT-116 cells) when compared to the 3D Stroma μTs, in which the FAPα signal was displayed in the fibroblast cytoplasm, particularly centered around the nucleus (Figure 7K,L, white arrows). Interestingly, we found that the FAPα signal showed cytoplasmic localization and also a nuclear one in both the fibroblasts and HCT-116 cells of the 3D CRC μTs (Figure 7H,I, white arrows for fibroblasts, yellow arrows for HCT-116 cells). In order to further investigate the transition of the NFs into CAFs, we explored Yes-associated protein 1 (YAP1) expression in the 3D CRC μTs (Figure 7M–O) and the 3D stroma μTs (Figure 7P–R) by immunofluorescence. The confocal images showed a high nuclear expression of YAP-1 in the fibroblasts of the 3D CRC μTs (Figure 7N,O, white arrows) when compared to the fibroblasts in the 3D stroma μTs, in which we detected a cytoplasmic signal of YAP-1 (Figure 7Q,R, white arrows). Furthermore, a nuclear signal of YAP-1 was also detected in the HCT-116 cells (Figure 7O, yellow arrows).

### 2.4. 5-Fluorouracil and Curcumin-Loaded Nanoemulsions Treatments on 3D CRC μTs

In order to explore the cytotoxic effect of the chemotherapeutic agent 5-FU in combination with Curcumin-loaded Nanoemulsions, we treated the 3D CRC μTs with different concentrations of 5-FU for 24 h, 48 h, and 72 h with or without CT-NE-Curc. The 3D stroma μTs and 3D HCT-116 μT monocultures were used as the controls. The graphs show a strong reduction in cell viability in the 3D CRC μTs in a dose-dependent manner after 24 h, 48 h, and 72 h of 5-FU treatment (Figure 8A–C, orange bars). In contrast, the 3D Stroma μTs showed a slight reduction in cell viability (Figure 8A–C, blue bars). Cell viability in the 3D CRC μTs was significantly reduced after 48 h and 72 h, resulting in more than 50% cell mortality, which is in contrast to the cell viability of the 3D Stroma μTs, where cell mortality always stayed above 50%. In contrast, Appendix A showed that in the in vitro 2D cultures, the selectivity of 5-FU was less evident between the HCT-116 cells and the less invasive colon cancer cells (Caco2). Furthermore, the 3D HCT-116 μTs were treated with the same dosages of 5-FU to observe how the cancer cells responded to the cytotoxic effect of the chemotherapeutic agent. The results revealed that the 3D HCT-116 μTs had good cell viability after 24 h, 48 h, and 72 h (Figure 8A–C, light blue bars) when compared to the 3D CRC μTs and 3D Stroma μTs, except for 1mM and 10 mM, for which the cancer cells in the 3D HCT-116 μTs showed a decrease in cell viability after 72 h of treatment (23.9 ± 0.3 and 15.3 ± 3.7). Thus, the 3D HCT-116 μTs are less responsive to 5-FU when compared to the 3D CRC μTs and the 3D Stroma μTs.

The combined treatment with CT-NE-Curc and 5-FU was carried out on the 3D CRC μTs, 3D Stroma μTs, and the 3D HCT-116 μTs. The graphs show that a cytotoxic effect of CT-NE-Curc/cell medium (with no 5-FU) towards the 3D CRC μTs was not observed after 24 h (81.1 ± 14.6), but it appears after 48 h of treatment (67.2 ± 3.2) (Figure 8D,E). As in the case of 5-FU, CT-NE-Curc/cell medium showed selective behavior for all the times and was particularly interesting after 48 h, when it demonstrated a nontoxic behavior towards the 3D Stroma μTs (96.6 ± 1.2) but was cytotoxic for the 3D CRC μTs (Figure 8E). Regarding the combination of CT-NE-Curc/5-FU, there was not a significant difference when compared to the corresponding treatments made with only 5-FU (Figure 8D–F). Additionally, we observed that were no critical differences between the 5-FU alone treatments and the combinations with CT-NE-Curc (also for the 3D CRC μTs), which is different from the in vitro 2D cell cultures where the combination of the two treatments showed enhanced results and a protective effect of the curcumin toward the healthy cells (Appendix A).

## 3. Discussion

In this study, we developed in vitro 3D CRC μTs using the porous microbead-based approach to faithfully reproduce the TME and explore the several phenomena underlying cancer development and progression.

Firstly, we produced several configurations of 3D CRC μTs and investigated different behaviors in terms of ECM remodeling and the dynamic bidirectional crosstalk between cancer cells and the TME. In the last years, several studies have focused on the influence of cancer cells in remodeling the surrounding ECM according to cancer cell invasion [19]. Indeed, invasive cancers often show a stromal desmoplastic reaction, a fibrotic condition characterized by increased synthesis and the altered organization of the collagen network, which is associated with a poor prognosis [20,21]. In this regard, we observed that the 3D CRC Day0 µTs saw a drastic reduction in collagen fibers, as reported in the in vivo studies [22], which is different from the 3D CRC Day4 µTs. The latter presented increased collagen fiber assembly, probably due to the presence of a neosynthetized ECM before the seeding of the HCT-116 cells that allow the TME complexity and uniform spatial cellular distribution to be reproduced, as shown in solid tumors in vivo [23]. On the contrary, the low level of neosynthetized ECM in the 3D CRC Day0 µTs could be due to the time point at which the HCT-116 cells were seeded. Since the HCT-116 cells are very invasive, their proteolytic activity has the upper hand on ECM production. As a result, we did not consider the 3D CRC Day0 µTs for further experiments due to a lack of a structured and organized ECM [21]. As an optimal one, we chose the configuration of the 3D CRC Day4 µTs due to the homogeneous spatial distribution of the HCT-116 cells in the endogenously produced ECM.

Another aspect to take into account is the ascorbic acid (AA) supplementation in the media. Many studies have investigated the relationship between AA addition in cell media and ECM deposition. Previous studies have observed that AA was stable in cell media, and its prolonged exposure to fibroblasts increased collagen deposition [24,25]. The concentration of AA (50 µg/mL) was previously proved to be effective for stimulating collagen synthesis in human fibroblasts [26]. According to the literature, we reported that the same concentration of AA does not interfere with the cancer cell viability in the 3D CRC Day4 µTs with AA [27], which also allowed for an increase in ECM production by day 12 of culturing. Since it is known that tumor ECM synthesis increases and collagen fibers are upregulated in several types of cancer [28], we hypothesized that the significant increase in the collagen synthesis by day 12 of culturing was due to cancer cells that act on continuous ECM remodeling to create a microenvironment that promotes tumorigenesis [29]. Then, we compared the 3D CRC Day4 µTs with AA to the 3D HCT-116 µTs and 3D Stroma µT monocultures, which were considered controls of cancer cell proliferation and ECM remodeling. In agreement with the literature, the collagen fibers are more organized and well-distributed in the healthy model than in the tumor model, in which the loss of fine structures and the structural organization of the collagen network were observed [30]. Interestingly, we found denser and newly synthesized collagen fibers in some of the regions of the 3D CRC Day4 µTs with AA. This aspect could be due to the continuous and altered ECM remodeling during cancer cell invasion through the native tumor stroma, characterized by a heterogeneous morphology and the organization of the collagen fibers [31]. Finally, we selected the 3D CRC Day4 µTs with AA configuration, named 3D CRC µTs, to investigate the ECM remodeling and bidirectional interplay between the cancer cells and stromal cells in a relevant TME.

It is known that cancer cells closely interact with their ECM, generating a bidirectional feedback loop with ongoing ECM remodeling. On the one hand, ECM stimulates cancer cell proliferation, migration, polarization, and cell–cell interactions. On the other hand, cancer cells alter the composition, structure, and mechanics of the ECM [32]. Additionally, the interaction between cancer cells and cancer-associated fibroblasts (CAF) promotes alterations in the collagen network by producing collagen fibers tracks, like “highways”, or by exerting mechanical pulling forces on collagen fibers to enable the invasion of cancer cells in bulk [33]. In this regard, we found highly aligned and oriented collagen fibers in some regions in the 3D CRC µTs, which was different from the 3D Stroma µTs, in which a random distribution of collagen fibers was observed. According to earlier research, in our 3D CRC µTs, we found that cancer cells have the ability to change their shape and the ECM around them to achieve optimal motility. This feature allows cancer cells to invade and migrate following the direction of the local collagen fibers [34,35].

Recent studies have shown that during cancer cell invasion, both the production and the breakdown of the ECM components increase, resulting in a tumor-reactive stroma [36]. In this respect, we found a high level of ECM degradation in the histological cross-section of the 3D CRC µTs when compared to the 3D Stroma µTs. Moreover, we noted that some cancer cells presented a round shape on the surface of the 3D CRC µTs, and others presented a spindle shape within the degraded ECM areas. We hypothesized that those cancer cells were going through an epithelial-mesenchymal transition (EMT) process, which occurs during the early stages of cancer progression [37]. It is well recognized that the heightened secretion of ECM-degrading molecules, such as matrix metalloproteinases (MMPs), is what causes the high breakdown of ECM molecules, such as type-I collagen. During the bidirectional interplay between cancer and stromal cells in several tumors, MMPs are continuously generated and secreted, promoting the hallmarks of tumor progression, such as angiogenesis, invasion, and metastasis [38]. In more detail, the gelatinase MMP-9 is crucial for ECM degradation in tumor invasion, metastasis, and angiogenesis and is directly linked to a poor prognosis in CRC patients [39]. In accordance with the literature, we found that the MMP-9 signal was stronger in the 3D CRC μTs than in the 3D Stroma μTs, and it was particularly localized in both the fibroblast and cancer cell protrusions of the plasma membrane [40]. This result confirmed an increase in ECM remodeling in the 3D CRC μTs, showing the ability of cancer cells to break down a number of ECM macromolecules and aid in their invasion and migration through the surrounding tissue.

Over the past few years, a growing number of pieces of research have demonstrated that the activation of normal fibroblasts (NFs) into CAFs occurs during the bidirectional crosstalk between cancer cells and NFs, triggering several mechanisms that induce the malignant behavior of cancer cells [41]. α-smooth muscle actin (αSMA) and fibroblast activation protein α (FAPα) are the two stromal markers most frequently employed to identify CAFs. According to the literature [42], we detected a strong cytoplasmic signal for αSMA in the fibroblasts of the 3D CRC µTs, which was different from the fibroblasts in the 3D Stroma µTs. In addition, we found that the cancer cells also showed a strong αSMA signal, suggesting their ability to acquire a mesenchymal phenotype during progression and invasion and determine a poorer prognosis, as reported in other studies as well [43]. FAPα is a homodimeric integral membrane gelatinase of the serine protease family that is selectively expressed by CAFs at the stromal compartment. Moreover, this gelatinase has been identified in a number of epithelial cancer cell lines and is thought to have a significant role in controlling the development of the disease [44]. Previous studies confirmed that the upregulation of FAPα might enhance the migration and invasion of CRC cells and that its expression increased with the progression of tumor staging [45]. As reported in the literature, we found a diffused FAPα signal in both the fibroblasts and HCT-116 cells in the 3D CRC μTs when compared to the 3D Stroma µTs, in which a slight signal was localized around the nucleus. Strikingly, the FAPα signal was localized in both the cytoplasm and nucleus of the fibroblasts and HCT-116 cells in the 3D CRC µTs. The role of nuclear FAPα is still unclear, but it could be linked to the activation of several mechanisms underlying cancer cell invasion and migration. This evidence is in line with the results of a previous study in which the nuclear signal of FAPα was found in CAFs [46]. Yes-associated protein 1 (YAP-1) is another important player in the TME and has a key role in regulating cell invasion, migration, survival, and the TME during metastasis [47]. Moreover, it is known that YAP-1 is upregulated within the nucleus of cancer epithelial cells but its activation in the transition of NFs into CAFs is not yet clear. According to the previous studies [48,49], we found a strong nuclear YAP-1 signal coming from the fibroblasts in the 3D CRC µTs when compared to the fibroblasts in the 3D Stroma µTs, in which a YAP-1 signal was localized within the cytoplasm.

Therefore, our 3D CRC μTs may represent promising and appropriate drug screening platforms. Indeed, 3D healthy and tumor tissue models are notoriously more representative than in vitro 2D cell culture models for the screening of drugs and drug delivery systems. In particular, we investigated the cytotoxic effect of the chemotherapeutic agent 5-FU alone and in combination with (CT-NE-Curc) in the 3D CRC µTs, 3D Stroma µTs, and the 3D HCT-116 µTs. In contrast to the treatments with 5-FU alone on the 2D Caco2 culture and 2D HCT-116 culture, we observed the selective cytotoxic impact of 5-FU on cancer cells in the 3D CRC µTs vs. the 3D Stroma µTs, as was reported in previous studies [50]. However, we did not observe a protective role for curcumin when CT-NE-Curc was used in combination with 5-FU. We hypothesized that the presence of a complex ECM in the 3D Stroma μTs and 3D CRC µTs could act as a barrier against nanoemulsions penetration [51]. In contrast, in our 2D cell cultures, which did not present ECM deposition at all, the protective effect of curcumin in the combination treatments was evident [52]. However, we found that the 3D Stroma μTs treated with the 5-FU and CT-NE-Curc combination treatments were more viable than 3D CRC, in which there was a rapid ECM turnover due to the proteases secreted by the cancer cells, making the matrix more permissive to nanocarrier infiltration. In this scenario, we hypothesize that the delayed action of the nanocarrier in the 3D μTs was due to the nanocarrier size (around 100 nm), which is much larger than a molecule such as 5-FU, thus taking more time to penetrate the μTs. On the other hand, the high cell viability in the 3D HCT-116 μTs treated with 5-FU alone and in combination with CT-NE-Curc could also be due to the microtissue structure and organization. Indeed, it is known that cancer cells are more proliferative and grow over each other in an uncontrolled manner, even when in contact with neighboring cells. This property, known as “contact inhibition of proliferation”, allows the cancer cells to form tighter junctions, which can block the diffusion of the chemotherapeutic agent to the outer layers [53,54]. In summary, we suggest that 3D HCT-116 μTs are not able to recapitulate the complexity of in vivo tumors because of the lack of interplay with the stromal component, which is not a physiological condition. Indeed, it is well known that in vivo tumors are not merely an aggregation of cancer cells but represent a complex entity in which cancer cells interplay with stroma cells and ECMs, and together, this contributes to cancer progression [55]. In contrast to other 3D tumor models based on extremely simplified structures and organizations of the TME [15], we suggested that our in vitro 3D models reproduce native tumor tissue with a more complex and structured microenvironment, which does not allow for the easy penetration of nanoemulsions through the matrix. In this perspective, the use of an in vitro 2D cell culture can help our understanding if an activity is missing due to a diffusion issue or to the real lack of activity. In the first case, one may work on the design of the nanocarrier to enhance the penetration properties in terms of antifouling or, for instance, size with multistage approaches. Despite promising results, we also recognize some of the limitations of our in vitro tumor models. The normal fibroblasts used to develop the 3D CRC μTs are not tissue-specific, and these could be a starting point to fabricate 3D CRC models with human primary intestinal fibroblasts, reproducing (even more faithfully) the native tissue. Further studies could be performed to validate our 3D tumor models by comparing them with 3D models based on the solutions/gels of the several protein extracts derived from animal and human tumor tissue with that in the existing literature [56,57]. Nevertheless, the proposed 3D CRC μTs could be a representative and useful platform when combined with tissue-on-chip technology aiming at addressing studies in cancer progression and drug discovery.

## 4. Materials and Methods

### 4.1. Cell Culture

The human colorectal carcinoma cell line, HCT-116 cells, transfected with pLVX-ZsGreen1-N1 (λ_ex_ 493 nm, λ_em_ 505 nm) viral vector were purchased from Clontech (San Jose, CA, USA), as previously reported [12], and were cultured in Dulbecco’s Modified Eagle’s Medium (DMEM, Himedia, Einhausen, Germany) supplemented with 10% Fetal Bovine Serum (FBS) (Merck-Sigma Aldrich, Milan, Italy), 200 mM L-Glutamine (Himedia, Einhausen, Germany), and 100 IU mL^−1^ Streptomycin/Penicillin (Himedia Einhausen, Germany), incubated at 37 °C in 5% CO_2_. Primary normal human dermal fibroblasts (NFs) extracted from healthy biopsies were grown in Eagle’s Minimum Essential Medium (EMEM, Himedia, Einhausen, Germany), supplemented with 20% FBS (Merck-Sigma Aldrich, Milan, Italy), 200 mM L-Glutamine (Sigma Aldrich), 100 IU mL^−1^ Streptomycin/Penicillin (Himedia, Einhausen, Germany), 100X Non-Essential Amino Acid (Euroclone, Milan, Italy) and incubated at 37 °C in 5% CO_2_.

### 4.2. In Vitro 3D Colorectal Cancer Microtissues Fabrication

As shown in Figure 8, the 3D microtissues (µTs) were fabricated via a bottom-up method using gelatin porous microbeads (GPMs), which were produced according to a modified double emulsion technique (O/W/O), as previously described [58]. Three different 3D µTs were produced: 3D co-culture of colorectal cancer microtissues (3D CRC µTs), consisting of HCT-116 cells and NFs; 3D colorectal cancer microtissues, produced by HCT-116 cells (3D HCT-116 µTs, monoculture), and 3D stroma microtissues, consisting of NFs (3D Stroma µTs, monoculture). In addition, two types of 3D CRC µTs were produced: 3D CRC Day0 µTs (HCT-116 and NFs were seeded together on the GPMs at day 0 of culture); 3D CRC Day4 µTs (NFs are seeded on the GPMs at day 0 and then, HCT-116 were seeded on 3D Stroma µTs at day 4 of culture). All microtissues were cultured in dynamic conditions in spinner flask bioreactors and were monitored for 12 days and analyzed at different time points. The experimental setup is illustrated in Appendix A. To produce the 3D CRC Day0 µTs and the 3D CRC Day4 µTs, 24 mg of GPMs were mixed together with HCT-116 cells/NF (1:2 cell ratio) at day 0 of culture or with NFs alone at day 0 of culture to produce 3D Stroma µTs, respectively. To promote cell seeding on GPMs, an intermittent stirring regime (30 min at 0 rpm; 5 min at 30 rpm) for 6 h was carried out. Then, dynamic cultures were kept under continuous stirring at 30 rpm for up to 12 days. Moreover, all 3D CRC µTs produced were grown in Dulbecco’s Modified Eagle’s Medium (DMEM, Himedia, Einhausen, Germany) supplemented with 10% FBS (Merck-Sigma Aldrich, Milan, Italy), 200 mM L-Glutamine (Himedia, Einhausen, Germany), and 100 IU ml^−1^ Streptomycin/Penicillin (Himedia, Einhausen, Germany). In the following days, the medium was changed every two days by adding 50 µg/mL of ascorbic acid (AA). The 3D CRC Day0 µTs were cultured in dynamic conditions from day 0 to day 12 of culturing. Instead, to produce the 3D CRC Day4 µTs, HCT-116 cells were seeded on the 3D Stroma µTs on day 4 of culturing, and an intermittent stirring regime (30 min at 0 rpm; 5 min at 30 rpm) was carried out for 6 h. Then, the dynamic cultures were kept under continuous stirring at 30 rpm for up to 12 days. Medium supplemented with (w) or without (w/o) AA was added on day 4 at the time of seeding the HCT-116 cells to produce two configurations of the 3D CRC Day4 µTs: 3D CRC Day4 w AA µTs and 3D CRC Day4 w/o AA µTs to see the different fibroblasts’ ECM production. Two 3D CRC Day4 µT types were cultured in dynamic conditions until day 12 of culturing. To fabricate the 3D Stroma µTs and 3D HCT-116 µTs, 24 mg of sterile GPMs were loaded with 1.2 × 10^6^ HCT-116 cells (10 cell/microbeads ratio) or 2.4 × 10^6^ NFs (20 cell/microbeads ratio), respectively. To promote NFs or HCT-116 cells seeding on GPMs, an intermittent stirring regime (30 min at 0 rpm; 5 min at 30 rpm) was carried out for 6 h. Then, the 3D Stroma µTs were kept under continuous stirring (dynamic condition) at 80 rpm for up to 12 Days. On the contrary, the 3D HCT-116 µTs were left in a static condition overnight. The culture medium used for the 3D Stroma µTs was Eagle’s Minimum Essential Medium (EMEM, Himedia, Einhausen, Germany) supplemented with 20% Fetal Bovine Serum (FBS, Sigma Aldrich), 200 mM L-Glutamine (Merck-Sigma Aldrich, Milan, Italy), 100 IU ml^−1^ Streptomycin/Penicillin (Himedia, Einhausen, Germany), and 100X Non-Essential Amino Acid (Euroclone, Milan, Italy). For the 3D HCT-116 µTs, Dulbecco’s Modified Eagle’s Medium (DMEM, Himedia, Einhausen, Germany) supplemented with 10% FBS (Sigma Aldrich), 200 mM L-Glutamine (Himedia, Einhausen, Germany), and 100 IU mL^−1^ Streptomycin/Penicillin (Himedia, Einhausen, Germany) was used. For the 3D Stroma µTs, 50 µg/mL of AA was added to the culture medium every two days until day 12 of culturing. All microtissues were maintained at 37 °C in a humidified 5% CO_2_ incubator. All the 3D µTs were collected at different time points and processed for cell proliferation and ECM remodeling analyses. Instead, the histological, immunofluorescent analyses and combination treatments were carried out for all of the 3D µTs at day 12 of culturing.

### 4.3. Colorectal Cancer Cells Counting

To evaluate the tumor cell proliferation in the 3D CRC µTs, 3D CRC µTs, and the 3D HCT-116 µTs, aliquots (500 µL) were collected during microtissue assembling time on days 4, 8, and 12, and these were then fixed with 4% paraformaldehyde (PFA). After that, the images were acquired by Confocal Leica TCS SP5 II (Leica Microsystems, Milan, Italy), where the 3D HCT-116 µTs and CRC µTs were observed by using a laser that excited green fluorescent protein (GFP), expressed by the HCT-116 cells (λ_ex_ = 488 nm and λ_em_ = 510 nm). Colorectal cancer cell nuclei showed GFP expression and were counted by using the Fiji plugin “Cell Counter”. In detail, 4 images were selected for each time point, and the counting was carried out by selecting 5 Region of Interest (ROI) (100 × 100 µm^2^) in each image. The results of cell counting were expressed as the ratio between the total cell number and ROI area (cell number/µm^2^).

### 4.4. Collagen Fraction and Degree of Collagen Assembly Analysis

In order to analyze collagen production and assembly degree, the samples were observed under Confocal Leica TCS SP5 II combined with a Multiphoton Microscope (Leica Microsystems, Milan, Italy), where the NIR femtosecond laser beam was derived from a tunable compact mode-locked titanium: sapphire laser (Chamaleon Com-pact OPO-Vis, Coherent). Unstained collagen of the samples was observed (SHG, λ_ex_ = 840 nm, λ_em_ = 420 ± 5 nm). Then, the 3D CRC µT images acquired by the SHG technique were analyzed by using Fiji software. Collagen fraction analysis was carried out by measuring the collagen portion in the ECM space that corresponds to bright pixels (number of pixels from the collagen—Nc), with respect to black pixels, which represent the noncollagen portion (number of pixels noncollagen portion—Nb). For each time point, the collagen fraction was expressed as the ratio between bright pixels (Nc) and the total of the bright pixels and the black pixels (Nb) in the selected ROI, as reported in the following Equation:$$Collagen\;Fraction=\frac{Nc}{Nc+Nb}$$

Moreover, the degree of collagen assembly (CAD) was evaluated by analyzing the intensity of the SHG signal. The SHG images were analyzed in order to calculate the average intensity, as described by the following Equation:$$CA\propto \underline{I}=\frac{\sum_{i=1}^{255}I_{i}p_{i}}{\sum_{i=1}^{255}p_{i}}$$
where $\underline{I}$ is the average intensity, $I_{i}$ is the intensity corresponding to the pixel $p_{i}$, while the index *i* = *xi*, *yi* runs in the gray value interval from 1 to 255. The intensity $\underline{I}$ of the collagen network is known to be proportional to both the mechanical properties and the degree of assembly of the newly synthesized collagen.

### 4.5. Collagen Fiber Orientation and Alignment Analysis

To determine the changes in the orientation of individual collagen fibers in the 3D CRC µTs and the Stroma µTs, a quantitative analysis of collagen fiber orientation was performed by using the Fiji plugin “Orientation J”. The plugin determines the local orientation, energy, and coherency for every pixel in the image [59]. It produces a hue-saturation-brightness (HSB) color-coded map, which shows the angles of the collagen fibers orientation in the image. The following parameters were used: Gaussian window σ 1 pixel, Min Coherency 20%, and Min Energy 2%. In addition, we measured the coherency index (COI), which indicates the degree to which the collagen fibers were oriented. The eccentricity of the ellipses demonstrates the coherency value: the narrower the ellipse indicates a higher coherency value (=1), in which the collagen fibers are fully aligned, while a perfect circle indicates a lower coherency value (=0), in which the collagen fibers have a completely random distribution. The plugin calculates the distribution of the fiber alignment directions comparing the HCT-116 cells orientation in degree.

### 4.6. Histological and Immunofluorescence Analyses in 3D Colorectal Cancer Microtissues and 3D Stroma Microtissues

For the histological and immunofluorescence analyses, both 3D CRC µTs and 3D Stroma µTs were fixed in 4% PFA. In detail, for the histological analysis, the samples were dehydrated in Ethanol from 75% to 100% and treated with Xylene (A9982 ROMIL, Naples, Italy) before paraffin inclusion. Microtissue slices 7µm-thick were cut using a microtome, laid in warm water, and left in the oven at 30–40 °C to dry overnight. Then, the sections were deparaffinized using xylene, hydrated in ethanol from 100% to 75%, washed in water, and stained using Hematoxylin/Eosin (H/E) (Bio-Optica, Milan, Italy)). The sections were mounted with Histomount Mounting Solution (Bio-Optica, Milan, Italy) on coverslips, and the morphological features of μTs were observed with a light microscope (Olympus, BX53, Milan, Italy). For immunofluorescence analysis, 3D CRC µTs and 3D Stroma µTs were withdrawn from the dynamic culture at the end of culture time, washed twice with Phosphate buffered saline (PBS, Euroclone, Milan, Italy), and then fixed with 4% PFA for 20 min. After washing, the samples were incubated with the permeabilizing solution (PBS-Triton X-100 0.1%) for 10 min, blocked with PBS-BSA 3% and 1% solutions, and incubated with primary antibody MMP-9 (ab119906, Abcam, Milan, Italy), α-SMA 1:100 (5694, Abcam, Milan, Italy), FAPα (PA5-99313, Invitrogen, Milan, Italy), and YAP-1 1:250 (PA1-46189, Invitrogen, Milan, Italy) for 2 hours at Room Temperature (RT), respectively. Then, the samples were incubated with secondary antibody 1:500 (Alexa fluor 546, Invitrogen, Milan, Italy) for 1 h at RT, and the nuclear stain was performed by applying a diluted suspension (1:1000) of 1, 5-bis-4, 8-dihydroxyanthracene-9, 10-dione (DRAQ-5, Invitrogen, Milan, Italy) for 15 min at RT. The images were obtained using confocal laser scanning microscopy (CLSM, Leica Microsystems, Milan, Italy). The samples were observed to highlight the simultaneous excitation of the two different fluorophores used (DRAQ5 λ_ex_ = 488/647 nm, λ_em_ = 647/681 nm; α-SMA and YAP-1 λ_ex_ 540–545 nm, λ_em_ 570–573 nm).

### 4.7. Curcumin-Loaded Nanoemulsions Preparation 

The nanoemulsion consisted of an oil-in-water (O/W) core coated with a mucoadhesive chitosan derivative, namely chitosan. The oil-in-water nanoemulsion was obtained by using a high-pressure homogenizer (Microfluidics M110PS, Massachusetts, USA), as previously described [16,17,18]. Briefly, 5.8 g of surfactant (egg lecithin) in 24 mL of oil (soybean oil) was used, and the oil phase was loaded with 100 mg of curcumin. To promote dissolution, the oil phase containing the surfactant and curcumin was mixed by alternating a high-speed blender (RZR 2102 control, Heidolph, Merck-Sigma Aldrich, Milan, Italy) at 60 °C and 500 rpm to RT sonication with an immersion sonicator (Ultrasonic Processor VCX500 Sonic and Materials, Newtown, USA), according to a process protocol previously reported. Then, to obtain the pre-emulsion, the oil phase was added dropwise to 90 g of Milli-Q water and mixed using the immersion sonicator under temperature control in order to avoid overheating. The pre-emulsions were finally passed at 2000 bar through the high-pressure valve homogenizer (Microfluidics M110PS, Massachusetts, USA), according to the same previous protocol. The primary nanoemulsion with curcumin loaded was then coated with chitosan, exploiting their positive charge for the deposition around the oil droplets stabilized with lecithin, which is negative. Chitosan solution was prepared in 0.1 M acetic acid Milli-Q water, bringing the pH to 4 with a small addition of NaOH 6 M solution. The two phases were mixed 1:1 (v:v) quickly under vigorous stirring and kept under stirring for 15 min to allow uniform chitosan deposition, thus obtaining secondary emulsions. The final concentration of oil was 10 wt%, whereas chitosan was 0.1 wt% to guarantee the saturation of chitosan around the emulsion. The emulsions coated with chitosan were loaded in a high-pressure valve homogenizer at 700 bar for 100 continuous steps and reprocessed after a few days in the same conditions using the same systems (at 700 bar for 100 continuous steps), having found a benefit in terms of stability from the double redispersion process. Nanoemulsions exhibited a size of 94.87 nm in diameter, a Z-potential of +21.5, and a polydispersity index (PDI) of 0.08.

### 4.8. Cytotoxicity Assessment of 5-Fluorouracil and Curcumin-Loaded Nanoemulsions on 3D Microtissues

For drug treatments cytotoxicity assessment, a stock solution of 5- Fluorouracil (5-FU) stock solution in Dimethyl sulfoxide (DMSO, Merck-Sigma Aldrich, Milan, Italy) [384 mM] was diluted in DMEM high glucose (Himedia, Einhausen, Germany) (10% FBS, 200 mM L-Glutamine, 100 IU ml^−1^ Streptomycin/Penicillin) at different concentrations: 0 μM; 10 μM; 100 μM; 1 mM; and 10 mM in DMSO at a concentration of 384 mM at different concentrations in DMEM high glucose supplemented with 10% FBS (Merck-Sigma Aldrich, Milan, Italy), 200 mM L-Glutamine (Himedia, Einhausen, Germany), 100 IU ml^−1^ Streptomycin/Penicillin (Himedia): 0 μM; 10 μM; 100 μM; 1 mM; and 10 mM. The final concentration of the DMSO was less than 1% of drug treatment 5-FU. Then, Curcumin-Loaded Nanoemulsions (CT-NE-Curc) was sterilized using PVDF filters (MerckMillipore, Milan, Italy) and diluted at 1:8 into the cell medium. Then, the 3D CRC µTs and 3D Stroma µTs were pre-treated with CT-NE-Curc and incubated at 37 °C in a humidified 5% CO_2_ for 2 h. After 2 h, the 3D Stroma µTs, 3D CRC µTs, and the 3D HCT-116 µTs were washed two times with sterile PBS and were treated with 5-FU at 0 μM; 10 μM; 100 μM; 1 mM; and 10 mM. The sample effects were collected, observed after 24 h, and 48 h, and 72 h, and 5-FU cytotoxicity was evaluated by using 3-(4,5-dimethylthiazol-2-yl)-2,5-diphenyltetrazolium bromide (MTT) assay according to the manufacturer’s instructions (Merck-Sigma Aldrich, Milan, Italy).

### 4.9. Cell Vitality Assay

To evaluate cell viability, 3-(4,5-dimethylthiazol-2-yl)-2,5-diphenyltetrazolium bromide (MTT) assay was used, according to the manufacturer’s instructions (Sigma Aldrich). Briefly, after 24 and 48 h of treatment, both (3D Stroma µTs, 3D CRC µTs, and 3D HCT-116 µTs) 3D µTs were washed two times with PBS, and 200 μL of the MTT solution (5 mg/mL) was added in each well, incubating at 37 °C in a humidified environment at 5% CO_2_ for 3 h in the dark. Then, the formazan precipitates MTT solution was removed from each well and the remaining crystals (formazan precipitates) were solubilized with 200 μL of DMSO, and the cells were incubated for an additional 30 min at 37 °C with gentle shaking. In the end, the optical density of each well sample was measured with a microplate spectrophotometer reader (EnSpire^®^, Perkin Elmer, Massachusetts, MA, USA) at 570 nm, and the cell viability (%) was calculated via the following Equation:$$Cell\;Viability\;\left(\%\right)=\frac{OD\;treated\;}{OD\;control}\times 100$$

### 4.10. Statistics

All experiments were repeated in independent studies (n ≥ 3), and all results were expressed as the mean ± mean standard error (SEM). For cancer cell counting analysis, 4 images were selected for each time point, and the counting was carried out by selecting 5 Regions of Interest (ROIs) from each image. The results of the cell counting were expressed as the ratio between the total cell number and the ROI area. For the collagen fraction and the collagen assembly degree (CAD) analyses, 4 images were selected for each time point, in which 5 ROI within each image were selected measuring the collagen portion in ECM space. The collagen fibers orientation analyses were carried out on 4 images on day 12 by measuring the alignment of collagen fibers in ECM with HCT-116 cells in the ROI. The mean of the three measurements collected in the different zones of each section was calculated, and the results were expressed as the ratio between the coherency index and the ROI area. In addition, the distribution of the collagen fibers orientation was normalized using the min–max method to obtain a normal distribution of the data. The cancer cell counting results were statistically analyzed by a nonparametric statistical test (Friedman test). The differences between two or more groups were evaluated using one-way analysis of variance (ANOVA), followed by the Tukey HSD post-test. *p*-values < 0.05 denoted statistically significant differences.

## 5. Conclusions

In this work, we developed a novel in vitro 3D CRC µTs that were able to faithfully reproduce the complexity of an in vivo TME. We observed that the ECM within the 3D CRC μTs was deregulated and modeled in the presence of cancer cells, favoring the formation of preferential routes for invasion and migration through the surrounding ECM. Moreover, the high expression of some stromal markers (αSMA and FAPα) and the high frequency of the nuclear localization of YAP-1 due to fibroblast activation in the 3D CRC μTs demonstrated that our in vitro models reproduced the transformation of normal fibroblasts into CAFs. Furthermore, these systems could be easily used for organ/tissue-on-chip technologies due to their small size. We used 3D CRC μTs to evaluate the effect of the combination of 5-FU and CT-NE-Curc. The results suggest that these models may be useful to evaluate the efficacy of novel drugs and/or nutraceuticals and DDSs based on a combination of activity and penetration ability, as well as the crucial crosstalk between cancer and stromal cells in the TME. However, the use of in vitro 2D models should not be overlooked since they allow us to understand if there are diffusion/penetration issues regarding nanocarriers or a lack of activity regarding the encapsulated molecule. The optimal design of nanocarrier properties is under development to enhance drug penetration and delivery in more complex in vitro models in order to translate the protective and synergistic effect of curcumin with 5-FU.

## Figures and Tables

**Figure 1 ijms-24-05678-f001:**
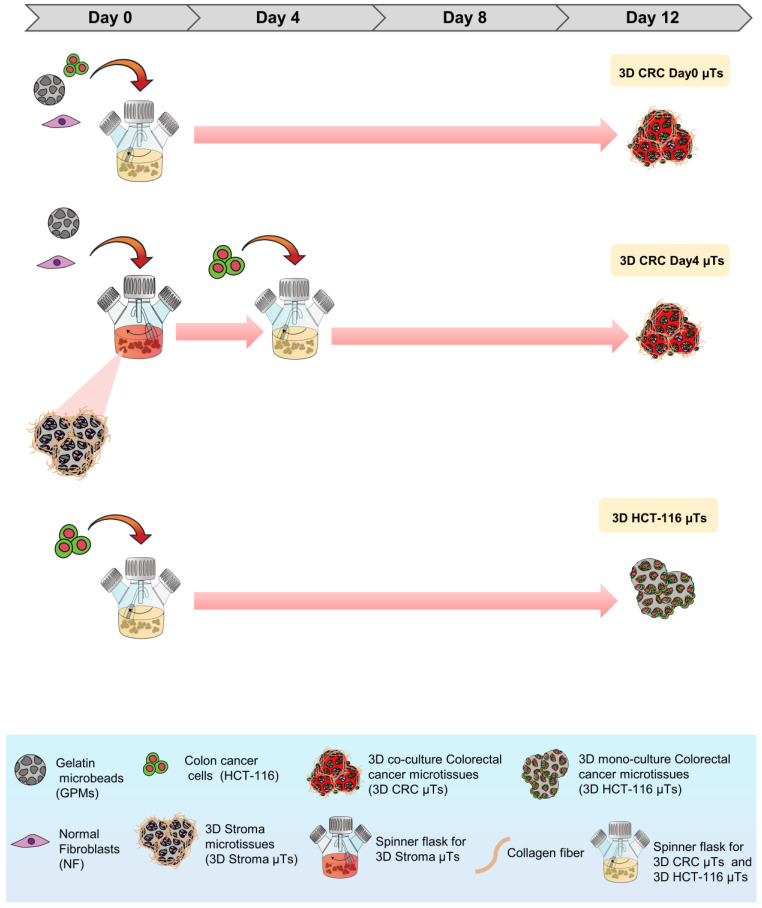
A schematic diagram of the method to produce the different 3D CRC µTs. GPMs, HCT-116 cells, and NFs were inserted into the spinner flasks to produce the 3D co-culture of colorectal cancer microtissues (the first consisted of HCT-116 cells, and NFs seeded together on day 0 of culturing, 3D CRC Day0 µTs, and the second consisted of HCT-116 cells seeded on 3D Stroma µTs on day 4 of culturing, 3D CRC Day4 µTs), and the 3D monoculture colorectal cancer microtissues (consisting of HCT-116 cells alone) (3D HCT-116 µTs) for 12 days.

**Figure 2 ijms-24-05678-f002:**
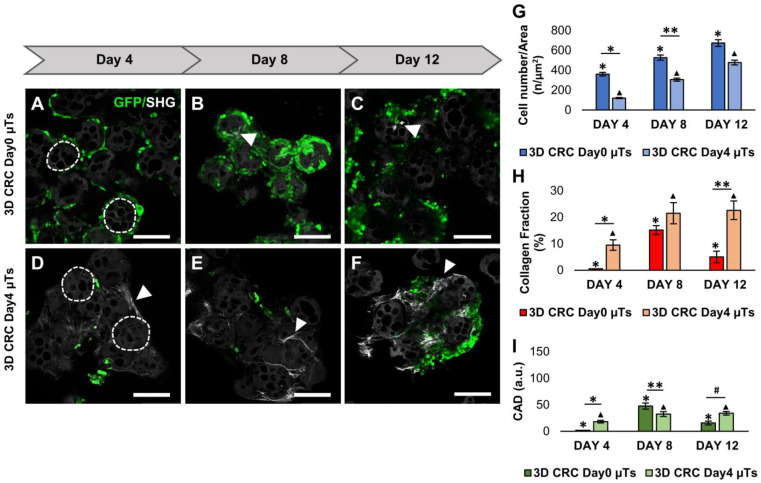
Comparison of cancer cell counts and extracellular matrix (ECM) remodeling between the 3D CRC Day0 µTs and 3D CRC Day4 µTs monitored on days 4, 8, and 12 of culturing. (**A**–**C**) Confocal images of 3D CRC Day0 µTs; (**D**–**F**) confocal images of 3D CRC Day4 µTs; scale bar: 150 µm. Gelatin microbeads are indicated by the dotted lines, green fluorescent protein (GFP) expressed by HCT-116 cells (green), and collagen Second Harmonic Generation (SHG) signal (white arrows). (**G**) Graph of cell numbers over time for the HCT-116 cells in the microtissues; (**H**) collagen fraction analysis; (**I**) collagen assembly degree (CAD) analysis. All the experiments were performed in triplicate (n = 3), and values represent the mean and the mean standard error (∗; ∗∗; ▲; # *p* < 0.05).

**Figure 3 ijms-24-05678-f003:**
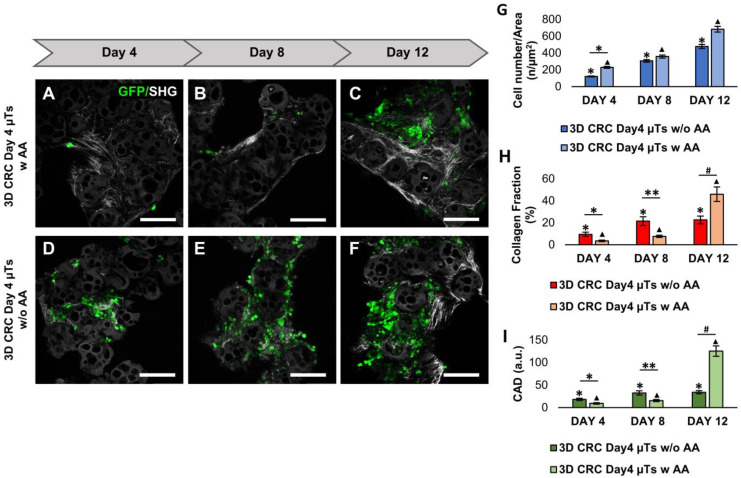
Comparison of cancer cell counts and extracellular matrix (ECM) remodeling between the 3D CRC Day4 µTs with Ascorbic Acid (AA) and 3D CRC Day4 µTs without AA monitored on days 4, 8, and 12 of culturing. (**A**–**C**) Confocal images of 3D CRC Day4 µTs w AA; (**D**–**F**) confocal images of 3D CRC Day4 µTs w/o AA; scale bar: 150 µm: green fluorescent protein (GFP) expressed by HCT-116 cells (green) and collagen Second Harmonic Generation (SHG)  signal (gray). (**G**) Graph of cell numbers over time for the HCT-116 cells in the microtissues; (**H**) collagen fraction analysis; (**I**) collagen assembly degree (CAD) analysis. All the experiments were performed in triplicate (n = 3), and the values represent the mean and the mean standard error (∗; ∗∗; ▲; # *p* < 0.05).

**Figure 4 ijms-24-05678-f004:**
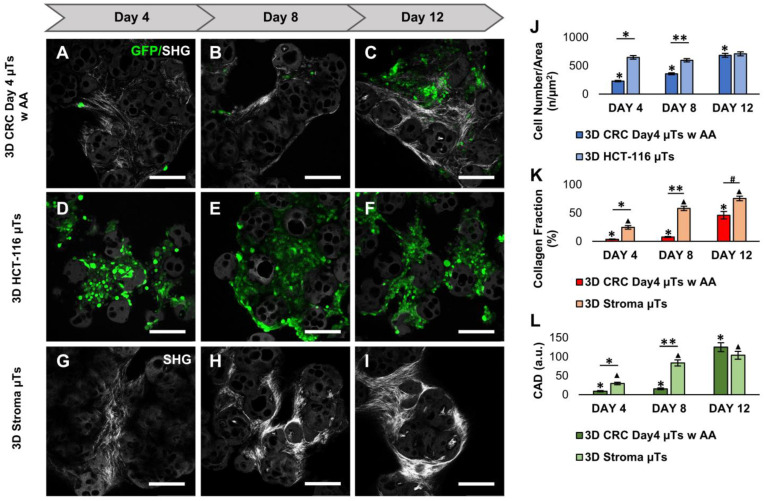
Comparison of cancer cell counts and extracellular matrix (ECM) remodeling between the 3D CRC Day4 µTs with Ascorbic Acid (AA), 3D HCT-116 µTs, and the 3D Stroma µTs, monitored on days 4, 8, and 12 of culturing. (**A**–**C**) Confocal images of the 3D CRC Day4 µTs with AA. (**D**–**F**) Confocal images of the 3D HCT-116 µTs. (**G**–**I**) Confocal images of the 3D Stroma µTs; scale bar: 150 µm; green fluorescent protein (GFP) expressed by HCT-116 cells (green) and collagen Second Harmonic Generation (SHG) signal (gray). (**J**) Graph of cell numbers over time for the HCT-116 cells in the 3D CRC Day4 µTs with AA and the 3D HCT-116 µTs; (**K**) Graph of collagen fraction analysis, and (**L**) collagen assembly degree (CAD) analysis for the 3D CRC Day4 µTs with AA and the 3D Stroma µTs. All the experiments were performed in triplicate (n = 3), and the values represented the mean and the mean standard error (∗; ∗∗; ▲; # *p* < 0.05).

**Figure 5 ijms-24-05678-f005:**
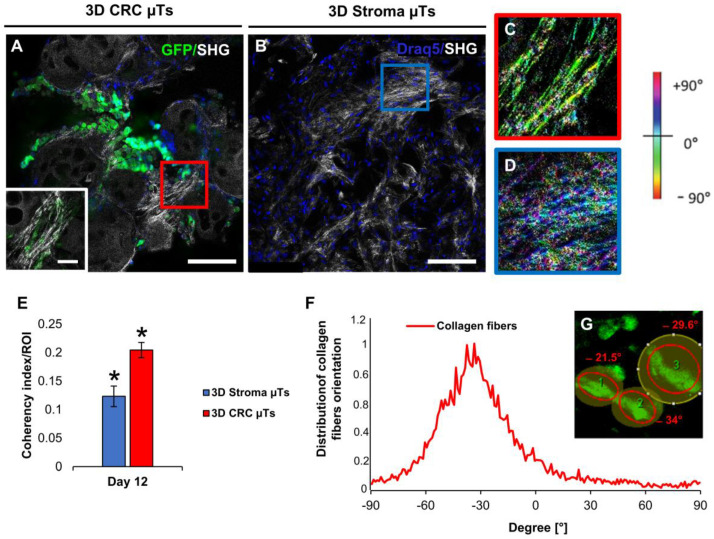
Collagen fiber orientation analysis of the 3D CRC μTs and 3D Stroma μTs on day 12 of culturing. (**A**,**B**) The confocal images of the 3D CRC μTs with a high magnification inset (scale bar 40 µm) and 3D Stroma μTs, respectively; green fluorescent protein (GFP) expressed by HCT-116 cells (green), nucleus of HCT-116 and Normal Fibroblasts/Cancer-Associated Fibroblasts (blue), and collagen Second Harmonic Generation (SHG) signal (gray); scale bar: 150 µm. (**C**,**D**) Color map of the collagen fibers organization in the 3D CRC μTs (red perimeter) and the 3D Stroma μTs (blue perimeter) insets; scale bar: 40 µm. (**E**) The coherency index/Region of Interest (ROI) graph of the 3D CRC μTs and 3D Stroma μTs. (**F**,**G**) Analysis of the distribution of collagen fibers orientation in the 3D CRC μTs. All the experiments were performed in triplicate (n = 3), and the values represent the mean and the mean standard error (∗ *p* < 0.05).

**Figure 6 ijms-24-05678-f006:**
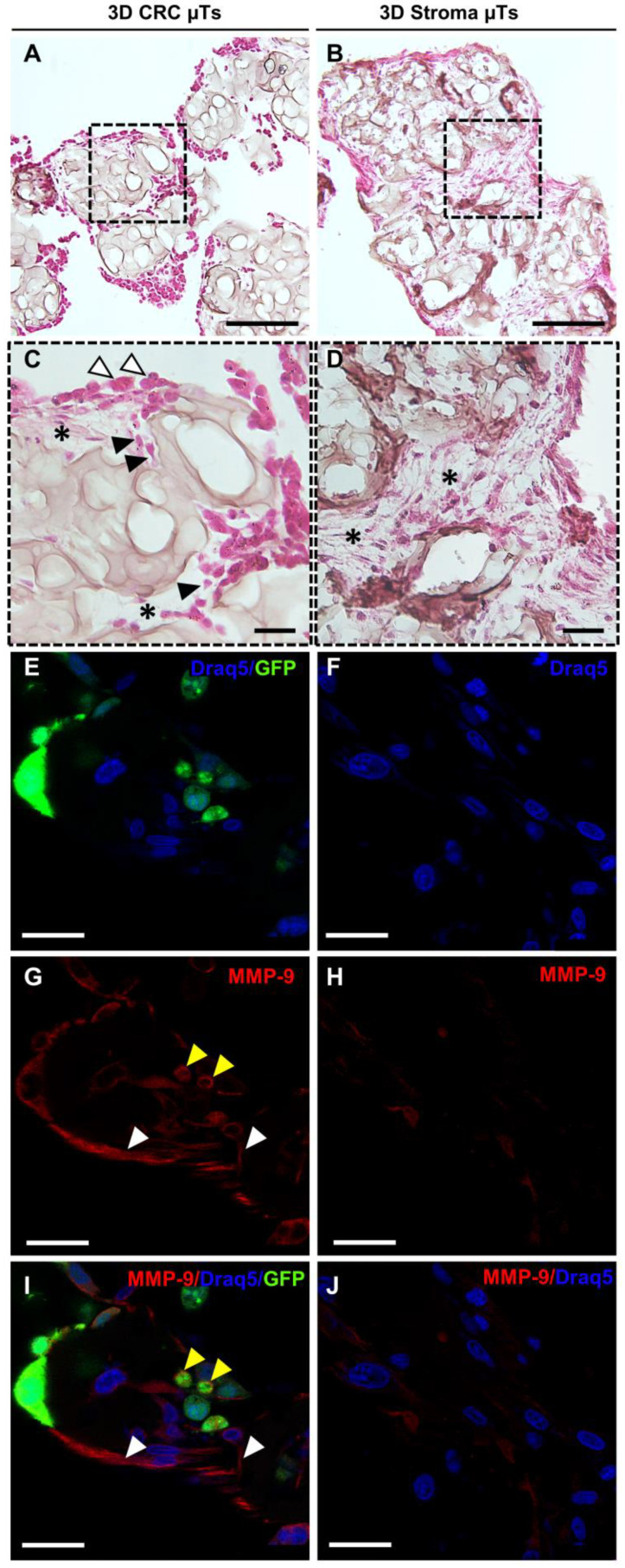
Extracellular Matrix (ECM) remodeling in the 3D CRC μTs and the 3D Stroma μTs at day 12 of culturing. (**A**,**B**) Histological section of the 3D CRC μTs and the 3D Stroma μTs, respectively, stained with H&E; scale bar: 100 μm. (**C**,**D**) Histological insets of the 3D CRC μTs and the 3D Stroma μTs with the presence of automatically produced ECM (indicated with asterisks), respectively, and I morphological changes in the HCT-116 cells, indicated by the white (round shape) and black (spindle shape) arrows; scale bar: 30 μm. (**E**,**G**,**I**) and (**F**,**H**,**J**): Immunofluorescence staining of the 3D CRC μTs (HCT-116 cells and fibroblasts indicated by yellow and white arrows, respectively) and the 3D Stroma μTs; scale bar: 50 μm. Nuclear staining with Draq5 (blue), green fluorescent protein (GFP) signal of HCT-116 cells (green), and Matrix metalloproteinase-9 (MMP-9) signal (red).

**Figure 7 ijms-24-05678-f007:**
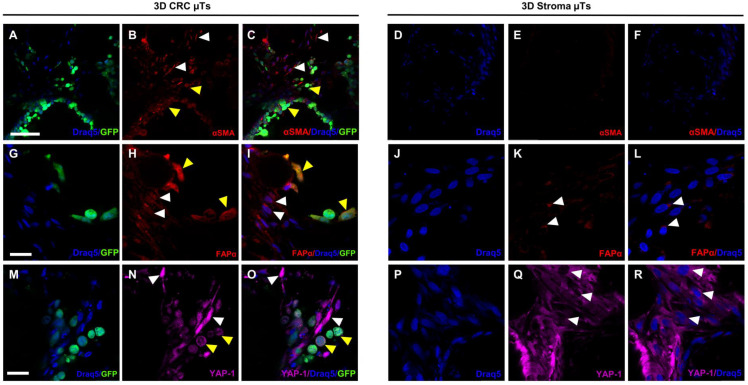
Immunofluorescence staining of the activation of Normal Fibroblasts (NFs) into Cancer-associated Fibroblasts (CAFs) in the 3D CRC μTs. (**A**–**C**) and (**D**–**F**): α-smooth muscle actin (αSMA) signal (red) in the 3D CRC μTs and the 3D Stroma μTs, respectively; scale bar: 100 µm. (**G**–**I**) and (**J**–**L**): Fibroblast Activation Protein α (FAPα) signal (red) in the 3D CRC μTs and the 3D Stroma μTs; scale bar: 50 µm. (**M**–**O**) and (**P**–**R**): Yes-associated protein 1 (YAP-1) signal (magenta) in the 3D CRC μTs and the 3D Stroma μTs; scale bar: 50 µm. CAFs and HCT-116 cells indicated by white and yellow arrows, respectively. Nuclear staining with Draq5 (blue) and green fluorescent protein (GFP) signal by HCT-116 cells (green).

**Figure 8 ijms-24-05678-f008:**
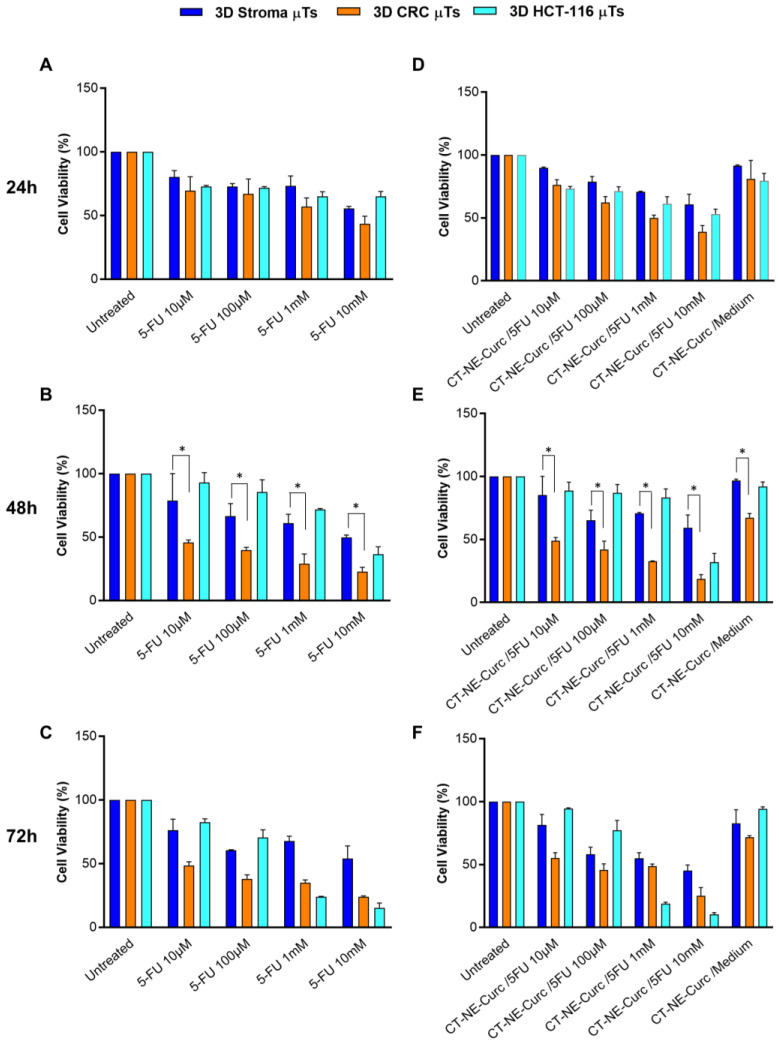
Cell viability assay for the 3D CRC μTs and 3D Stroma µTs treated with 5- Fluorouracil (5-FU) alone and in combination with Curcumin-loaded Nanoemulsions (CT-NE-Curc). (**A**–**C**) Graphs of the treatments with different concentrations of 5-FU at 24 h, 48 h, and 72 h; (**D**–**F**) graphs of the combination treatments with CT-NE-Curc and different concentrations of 5-FU at 24 h, 48 h, and 72 h. All the experiments were performed in triplicate (n = 3); the values represent the mean and mean standard error (∗ *p* < 0.05).

## Data Availability

Not applicable.

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
