# Peer review of "Colorectal Cancer Bioengineered Microtissues as a Model to Replicate Tumor-ECM Crosstalk and Assess Drug Delivery Systems In Vitro"

_ijms, 2023, doi:10.3390/ijms24065678_

Round 1

Reviewer 1 Report

Introduction: Please make separate paragraphs instead of one. Also, I  request write the aims and objectives of the study as a separate paragraph.

Result and Discussion: The section needs to be separated into two sections,  "Results" and "Discussion". 

At the end of the Discussions, please add a section on the strengths and limitations of the study, and another one on future directions. One of the limitations is, the study is based on Gelatin and collagen, which are not exact tumor microenvironment-mimicking matrices. 

In future direction: please add a section that this results should be further validated using 3D culture models where the cancer cells (https://doi.org/10.1016/j.yexcr.2018.06.037 ; https://doi.org/10.1186/s12885-015-1944-z ) was analyzed using animal and human-microenvironment based 3D tissue models. These matrices will further suit the human microenvironment-mimicking matrices. 

Author Response

We would like to thank the Reviewers for providing constructive and detailed review comments on our manuscript. The recommendations and advice have helped me/us to significantly enhance the quality of the manuscript. We have revised the manuscript according to their suggestions and addressed all their concerns.

We have revised our manuscript according to most of the reviewers’ comments.

The changes are marked up using the “Track Changes” function in the attached manuscript, and our point-by-point responses are indicated after the word “Author Response:”.

Comments and Suggestions for Authors

Reviewer 1

Introduction: Please make separate paragraphs instead of one. Also, I request write the aims and objectives of the study as a separate paragraph.

Author Response 1: As suggested by the reviewer, we separated the Introduction section in paragraphs in the revised manuscript. We also separated the study objectives as a separate paragraph.

Result and Discussion: The section needs to be separated into two sections,  "Results" and "Discussion".

Author Response 2: As suggested by the reviewer, we separated “Results” and “Discussion” in two sections in the revised manuscript.

At the end of the Discussions, please add a section on the strengths and limitations of the study, and another one on future directions. One of the limitations is, the study is based on Gelatin and collagen, which are not exact tumor microenvironment-mimicking matrices.

Author Response 3: We thank the Reviewer for his/her suggestion. We are confident that the Reviewer intended to suggest adding a paragraph on strengths and limitations, and another paragraph on future perspectives.  However, we added the limitations of this study and future directions at the end of the “Discussion” section in the revised manuscript.

Line 512-521: Despite promising results, we also recognize some limitations of our in vitro tumor models. The normal fibroblasts used to develop 3D CRC μTs are not tissue-specific, and these could be a starting point to fabricate 3D CRC models with human primary intestinal fibroblasts, reproducing even more faithfully the native tissue. Further studies could be performed to validate our 3D tumor models by comparing with 3D models based on solution/gel of the several proteins extracts derived from animal and human tumor tissue with that existing in the literature [56, 57]. Nevertheless, the proposed 3D CRC μTs could be a representative and useful platform to be combined with tissue-on-chip technology aiming at addressing studies in cancer progression and drug discovery.  

Regarding the limitation suggested kindly by the Reviewer about gelatin and collagen, we would like to explain better our approach. Our strategy for 3D tumor microtissues (3D CRC µTs) fabrication consists of producing a 3D engineered stroma compartment, guiding the cells to produce and self-assemble their own ECM molecules, including the collagen fibers. To do this, we used porous and biodegradable gelatin microbeads (GPMs) as scaffolds, where normal human fibroblasts are seeded within and induced to produce their own ECM by a dynamic culture condition. After 4 days of culture, we observed a complex organization and structure of stroma (3D Stroma µTs), and colon cancer cells (HCT-116) were seeded on 3D Stroma µTs for 3D CRC µTs production in order to investigate how cancer cells remodel a pre-existing and well-structured endogenous ECM towards a tumor ECM. Our previous studies have demonstrated that the GPMs provide suitable and mechanical support for adhesion, proliferation and synthesis of collagen and other ECM molecules by fibroblasts and the presence of a conspicuous ECM among and inside GPMs was found (https://doi.org/10.1016/j.actbio.2010.01.026). In summary, we used the GPMs not to mimic the TME matrix, but as a scaffold to let cells adhesion and due to the dynamic cell seeding process established, cells are able to synthesize and assemble collagen forming micotissue composed by GMP, cells and de novo synthesized ECM.

https://doi.org/10.1016/j.actbio.2010.01.026: Palmiero, C.; Imparato, G.; Urciuolo, F.; Netti, P., Engineered dermal equivalent tissue in vitro by assembly of microtissue precursors. Acta Biomaterialia 2010, 6, (7), 2548-2553.

In future direction: please add a section that this results should be further validated using 3D culture models where the cancer cells (https://doi.org/10.1016/j.yexcr.2018.06.037 ; https://doi.org/10.1186/s12885-015-1944-z ) was analyzed using animal and human-microenvironment based 3D tissue models. These matrices will further suit the human microenvironment-mimicking matrices.

Author Response 4: We thank the Revisor for pointing this out. We are confident that the Reviewer meant to suggest adding his/her comment as a paragraph on the future direction. Therefore we added the Reviewer’s suggestion in the paragraph on future directions at the end of the “Discussion” section. Moreover, we added also the references suggested kindly by the Reviewer.

Line 516-518: Moreover, our results will be further validated using 3D tumor models based on solution/gel of the several proteins extracts derived from animal and human tumor tissue in order to compare our TME with that reproduced by models existing in the literature [56,57]

Reviewer 2 Report

The manuscript is in an unusual format, to say the least; and probably has to be re-arranged if it is ever going too be accepted in a future version of the current manuscript. 

The merging of results and discussion is probably not along the accepted formats for this journal; and it certainly isnt easy to read.  It would be much more straightforward and easier to read if results and discussion would be 2 separate sections. It would also be helpful (in my opinion) if the materials & methods follows directly after introduction. 

I assume this is the format from some peprevious submission to another journal, in which this way of separating the chapters was mandatory. But its not really that helpful here and should be changed. 

Results: 

The stainings shown in figures 1-3 are not very detailed and I have a hard time distinguishing between tumor and stroma cells in these pictures. This only gets better in Fig. 4 (SHG images)and in Fig 5 (MMP stainings) and also the resolution in these last 2 figures are a lot better than Fig. 1-3.  It is difficult to ee the actual cells, and cell boundaries in Fig. 1-3 and also in 4; since no DNA counterstain (DAPI, Hoechst 33342, or Draq5) have been used. its not visible if there is any higher order cell aggregation or differentiation on the surface of the gelatin beads, and how this may or may not resembe the histologies seen in primary tumours. 

Stainings in Fig 5 could also be expanded, as we still dont see very clearly if tissue-like structures and of which architecture are formed in the microbeads. The stroal tissues appear relatively genuine and compare well with histological images, but I am not so sure what to think of the HCT-116 beads? Its also quite a stretch to speak of EMT in this circumstance, based on very very few indications, such as MMP9 expression and a bit of invasive behaviour. An EMT is an active process of transformation; while here we may simply be observing the normal invasive cell motility intrinsic to HCT-116 cells. 

The most conclusive and clearest result may be shown in Fig. 6; monitoring the activation of CAFs by nearby tumor cells. Even here, clarity of the microscopic images isnt optimal, which is probably due to the small size of the beads that are investigated. The YAP-1 stainings are standing a bit in isolation, as this protein may relate to activation of the HIPPO pathway which is then not any further investigated. Instead,there are again allegations made to EMT, which I am not sure if they are sustainable without any further functional investigations. 

The drug and nanoparticle treatments are interesting, and point to significant differences in the chemosensitivity of tumor/stroma co-cultures, compared to tumor and/ or stroma monocultures. Its just counterintuitive, and a bit unexpected since everyone would guess that tumor/stroma interactions will provide the "tissue" with more resistance to any drug treatments, or stress situations, in general. I do not think that this is entirely atributable to differences in penetration of drugs and NPs, especially since the stromal "tissue" will not form much barrier functions, and its not shown by any higher resolution images that this would be the case in HCT-116.

Generally, I thin that the entire results & discussion section needs to be cleaned up, and some of the poorly validated "findings" either have to go or they have to be better validated. The stainings arent really that great and could be replaced by some higher resolution images, if possible. The manuscript also has to be rearranged (as outlined above) and probably sortened to focus more on the few things that appear rally convincing and clear - such as the fibroblast activation. There is still not too much true novelty here as many of the findings are just isolated and are not further pursued in the necessary detail. 

Concerning the figures, I would make a number of recommendations:

1) Figure 8 describes how the microtissues are generated, what are the steps of "production" of these µTs and what happens next, and should probably be Figure 1 instead - simply to introduce the experimental system to the reader.

Still, the drawings in Fig 8 dont really explain how exactly are tumor cells and/or fibroblasts loaded into the system. What's the experimental setting, what cell and tissue culture devices, plates, or slides are used, also for microscopy, etc... none of this is totally clear to me and I doubt it would be to other readers.

Figure 1: preparation of the figures just isnt as professional as it should. There are many elements missing in the figure legends, such as complete declaration what are the units shown in 2G, H and I? Cell number/area... in what dimension? collagen content - in which measure precisely? what exactly is CAD? and what is CAD analysis ???

Fig. 2: The exact same issues also apply for the very similar Fig. 2 which lacks the same details. 

Fig.3: again, here the same issues with details that should be explained more thoroughly for the reader to be able to understand what exactly is measured 

Author Response

We would like to thank the Reviewers for providing constructive and detailed review comments on our manuscript. The recommendations and advice have helped me/us to significantly enhance the quality of the manuscript. We have revised the manuscript according to their suggestions and addressed all their concerns.

We have revised our manuscript according to most of the reviewers’ comments.

The changes are marked up using the “Track Changes” function in the attached manuscript, and our point-by-point responses are indicated after the word “Author Response:”.

The manuscript is in an unusual format, to say the least; and probably has to be re-arranged if it is ever going too be accepted in a future version of the current manuscript.

Author Response 1: We thank the Reviewer for pointing this out. We are confident that the Reviewer meant to suggest using the IMRaD format. Although we are used to organizing manuscripts following the IMRaD structure, we found that the instructions for the authors of the journal requested (and still request) the “Results” and “Discussion” sections should appear before the “Materials & Methods” section. Furthermore, having a glance at the papers published in the latest issues, we ascertained, to the best of our knowledge, no original articles recently published in IJMS have the IMRaD format. Consequently, although we are used to organizing manuscripts following the IMRaD structure, previous evidence compels us to not autonomously shift the “Results” and “Discussion” sections with the “Materials & Methods” section; We refer to the Editors/Editorial board for elucidations and, in case of their positive feedbacks, we will be ready to evaluate changes regarding the manuscript organization (namely, the use of the IMRaD format).

The merging of results and discussion is probably not along the accepted formats for this journal; and it certainly isnt easy to read.  It would be much more straightforward and easier to read if results and discussion would be 2 separate sections. It would also be helpful (in my opinion) if the materials & methods follows directly after introduction.

Author Response 2: We thank the Reviewer for his/her suggestion. We revised “Results” and “Discussion” into separate sections. Regarding the “Materials & Methods” section, as already clarify in Author response 1, although we are used to organizing manuscripts following the IMRaD structure, we found that the instructions for the authors of the journal requested (and still request) the “Results” and “Discussion” sections should appear before the “Materials & Methods” section.

I assume this is the format from some peprevious submission to another journal, in which this way of separating the chapters was mandatory. But its not really that helpful here and should be changed.

Author Response 3: We thank the Reviewer for this comment. We clarify the reasons for which the manuscript has not an IMRaD format. Once collected both the results and the larger part of the potential references, we have begun to seek for journals and special issues appropriate for submitting our work. After we faced internally each other, we decided to submit the manuscript to Special Issue "Recent Advance in 3D Cultures". Only at that time we started to mainly focus on the manuscript structure and, in turn, only at that time we discovered that the instructions for the authors of the journal requested (and still request) the “Results” and “Discussion” sections should appear before the “Materials & Methods” section, and, to the best of our knowledge, no original articles recently published in IJMS have the IMRaD format (as already reported in the Author Response 1 and 2).

Results:

The stainings shown in figures 1-3 are not very detailed and I have a hard time distinguishing between tumor and stroma cells in these pictures. This only gets better in Fig. 4 (SHG images)and in Fig 5 (MMP stainings) and also the resolution in these last 2 figures are a lot better than Fig. 1-3.  It is difficult to ee the actual cells, and cell boundaries in Fig. 1-3 and also in 4; since no DNA counterstain (DAPI, Hoechst 33342, or Draq5) have been used. its not visible if there is any higher order cell aggregation or differentiation on the surface of the gelatin beads, and how this may or may not resembe the histologies seen in primary tumours.

Author Response 4: We thank the Reviewer for his/her comment, and we clarify the reason why the stromal cells were not visible in these images. Our main purpose was to observe and follow the cancer cell’s proliferation and collagen fiber remodeling in the different 3D CRC µTs configurations over time. Therefore, we decided to show the images in which only the constitutive GFP-expressing colon cancer cells (HCT-116) were displayed in the 3D CRC µTs configurations in order to better emphasize the main differences in terms of cancer cell number during the 12 days of the dynamic culture. Moreover, we modified the resolution of figures 1-3 and 4 in the revised manuscript. Regarding the gelatin beads, we also clarify this aspect. We used the gelatin beads not as a matrix to mimic a tumor ECM, but as a mechanical support to seed cells and, thanks to the established dynamic process conditions, stimulate stromal cells to produce their own ECM. In our previous works, we have demonstrated that this approach succeeded in reproducing the main features of PDAC [12] and human breast cancer [54].

[12] Brancato, V.; Comunanza, V.; Imparato, G.; Corà, D.; Urciuolo, F.; Noghero, A.; Bussolino, F.; Netti, P. A., Bioengineered tumoral microtissues recapitulate desmoplastic reaction of pancreatic cancer. Acta Biomaterialia 2017, 49, 152-166.

[54] Brancato, V.; Gioiella, F.; Imparato, G.; Guarnieri, D.; Urciuolo, F.; Netti, P. A., 3D breast cancer microtissue reveals the role of tumor microenvironment on the transport and efficacy of free-doxorubicin in vitro. Acta Biomater 2018, 75, 200-212.

Stainings in Fig 5 could also be expanded, as we still dont see very clearly if tissue-like structures and of which architecture are formed in the microbeads. The stroal tissues appear relatively genuine and compare well with histological images, but I am not so sure what to think of the HCT-116 beads? Its also quite a stretch to speak of EMT in this circumstance, based on very very few indications, such as MMP9 expression and a bit of invasive behaviour. An EMT is an active process of transformation; while here we may simply be observing the normal invasive cell motility intrinsic to HCT-116 cells.

Author Response 5: We thank the Reviewer for his/her suggestion. We expanded the Fig.5 and added the asterisks where the stroma is shown in the revised manuscript. Regarding the invasive behavior of HCT-116 cells, we agree with the Reviewer that more indications are needed to demonstrate the presence of EMT process in the cancer cells. Indeed, we hypothesize that cancer cells had the EMT features during their invasion due to both the cell morphological changes and the high level of MMP9 signal, as modified in the revised manuscript:

Line 425-427: We hypothesised that those cancer cells were going through an Epithelial Mesenchymal Transition (EMT) process, which occurs during the early stages of cancer progression Moreover, it is true that HCT-116 are highly invasive cancer cells, and indeed we used this cancer cell line to better investigate and reproduce rapidly the local cancer cell invasion into a relevant ECM.

The most conclusive and clearest result may be shown in Fig. 6; monitoring the activation of CAFs by nearby tumor cells. Even here, clarity of the microscopic images isnt optimal, which is probably due to the small size of the beads that are investigated. The YAP-1 stainings are standing a bit in isolation, as this protein may relate to activation of the HIPPO pathway which is then not any further investigated. Instead,there are again allegations made to EMT, which I am not sure if they are sustainable without any further functional investigations.

Author Response 6: Thank the Reviewer for his/her comment. We agree with the Reviewer that the Fig. 6 displays the monitoring of the activation of Normal Fibroblasts (NF) into CAFs by nearby tumor cells. We are confident that the Reviewer meant to suggest enlarging the microscopic images of the Fig. 6 because they are small sizes. In this respect, we magnified the confocal images in the revised manuscript. Moreover, we would like to clarify that the beads are not visible in these confocal images because we focused on detecting different expressions of the molecules mostly involved in the transformation of NFs in CAFs. Regarding the YAP-1 staining, we agree with the Reviewer that YAP-1 is regulated negatively by the HIPPO pathway. However, it is known in the literature that this transcriptional co-activator is involved in CAFs transformation [48,49]. In this regard, the aim of our work was not to investigate the HIPPO pathway, but we focused on the YAP-1 expression as a marker to identify the presence of CAFs in our 3D tumor models. Regarding the allegation made to EMT, we modified the sentence in the “Discussion” section in the revised manuscript:

Line 463-470: YAP-1 is another important player in the TME and has a key role in regulating the cell invasion, migration, survival, and EMT during metastasis [47]. Moreover, it is known that YAP-1 is upregulated within the nucleus of cancer epithelial cells, but its activation in the transition of NFs into CAFs is not clear yet. According to the previous studies [48, 49], we found a strong nuclear YAP-1 signal coming from fibroblasts in 3D CRC µTs, compared to fibroblasts in 3D Stroma µTs in which YAP-1 signal was localized into the cytoplasm.

[48] Shen, T.; Li, Y.; Zhu, S.; Yu, J.; Zhang, B.; Chen, X.; Zhang, Z.; Ma, Y.; Niu, Y.; Shang, Z., YAP1 plays a key role of the conversion of normal fibroblasts into cancer-associated fibroblasts that contribute to prostate cancer progression. Journal of Experimental & Clinical Cancer Research 2020, 39, (1), 36.

[49] Naktubtim, C.; Payuhakrit, W.; Uttarawichien, T.; Hassametto, A.; Suwannalert, P., YAP, a novel target regulates F-actin rearrangement-associated CAFs transformation and promotes colorectal cancer cell progression. Biomedicine & Pharmacotherapy 2022, 155, 113757.

The drug and nanoparticle treatments are interesting, and point to significant differences in the chemosensitivity of tumor/stroma co-cultures, compared to tumor and/ or stroma monocultures. Its just counterintuitive, and a bit unexpected since everyone would guess that tumor/stroma interactions will provide the "tissue" with more resistance to any drug treatments, or stress situations, in general. I do not think that this is entirely atributable to differences in penetration of drugs and NPs, especially since the stromal "tissue" will not form much barrier functions, and its not shown by any higher resolution images that this would be the case in HCT-116.

Author Response 7: We thank the Reviewer for his/her interest in the drug and nanoparticle treatments. We have previously observed, that the chemosensitivity of tumor/stroma co-cultures compared to tumor and/or stroma monocultures was different in the case of breast cancer. In that study [54], we found that CAF/MCF7 µTs were more sensitive to doxorubicin than MCF7 µTs, due to the different physical barriers that hinder the diffusion of drugs into the two models. The CAF/MCF7 µTs were characterized by the presence of abundant ECM but the latter lacks at all in the MCF7 µTs. Although, matrix components could be considered a barrier to drug transport and notoriously cancer cells can develop drug resistance, the rapid ECM turnover due to the proteases secreted by the cancer cells, in CAF/MCF7 µTs allowed more drugs to infiltrate more freely inside the matrix structure, increasing DOX anti-cancer activity. We hypothesized that a similar condition occurred also in the case of 3D CRC µTs vs 3D HCT-116 µTs mono-culture. Therefore, the high cell viability in 3D HCT-116 μTs, treated with 5-FU alone and in combination with CT-NE-Curc, compared to 3D CRC µTs, could be explained by the difficulty of drug and nanoparticle to overcome the physical barriers of HCT-116 μTs that similarly to MCF7 µTs are featured by tight morphology that blocks the diffusion of drug to the outer layers impairing the penetration. The higher viability of 3D Stroma μTs compared to 3D CRC μTs could also be correlated to the more permeable tissue matrix in the second case. Indeed, we reported the high magnification images of 3D CRC μTs and 3D Stroma μTs in the Supplementary file (Figure S3) in order to better show the differences in 3D CRC μTs and 3D Stroma μTs in terms of organization, deposition, and structure of collagen fibers network. These images show the evident reduction of collagen fibers content in 3D CRC μTs differently from the 3D Stroma μTs in which the ECM was more structured and organized.

[54] Brancato, V.; Gioiella, F.; Imparato, G.; Guarnieri, D.; Urciuolo, F.; Netti, P. A., 3D breast cancer microtissue reveals the role of tumor microenvironment on the transport and efficacy of free-doxorubicin in vitro. Acta Biomater 2018, 75, 200-212.

Generally, I thin that the entire results & discussion section needs to be cleaned up, and some of the poorly validated "findings" either have to go or they have to be better validated. The stainings arent really that great and could be replaced by some higher resolution images, if possible. The manuscript also has to be rearranged (as outlined above) and probably sortened to focus more on the few things that appear rally convincing and clear - such as the fibroblast activation. There is still not too much true novelty here as many of the findings are just isolated and are not further pursued in the necessary detail.

Author Response 8: We thank the Reviewer for his/her point of view. We separated and revisited the “Results” and “Discussion” sections. We increased the resolution in all figures and rearranged the manuscript according to the guidelines for the Authors of the Special Issue "Recent Advance in 3D Cultures" of IJMS journal. Although we did not use a novel technology to produce the microtissues, we think that all our results are promising and may be a springboard to reproduce more relevant 3D CRC models using tissue-specific fibroblasts and investigate deeply, from a gene expression perspective, the TME of colon cancer. Regarding the drug and nanoemulsions treatments, the results are still preliminaries and further experimental evidence regarding the diffusive and penetration properties of the nanoemulsions will be carried out.

Concerning the figures, I would make a number of recommendations:

1) Figure 8 describes how the microtissues are generated, what are the steps of "production" of these µTs and what happens next, and should probably be Figure 1 instead - simply to introduce the experimental system to the reader.

Author Response 9: We thank the Reviewer for his/her suggestions. We added a new Figure in Supplementary Materials (Figure S3) in which in detail we explain the process of µTs production in different steps. Moreover, we introduced the experimental system in the first subsection of the “Results” section and shifted the Figure 8 as Figure 1:

Line 124-132: In this study, we developed three different configurations of 3D µTs using a cell-seeded microbeads approach and monitored their evolution during the 12 days of culture: 3D co-culture of Colorectal Cancer microtissues (3D CRC µTs), consisting of Normal fibroblasts (NFs) and HCT-116 cells; 3D mono-culture Colorectal Cancer microtissues, consisting of  HCT-116 cells alone (3D HCT-116 µTs) and 3D mono-culture Stroma microtissues, consisting of NFs alone (3D Stroma µTs). In detail, we produced two types of 3D CRC µTs: the first consisted of HCT-116 cells and NFs seeded on together at day 0 of culture (3D CRC Day0 µTs), the second consisted of HCT-116 cells seeded on 3D Stroma µTs at day 4 of culture (Figure 1).

Still, the drawings in Fig 8 dont really explain how exactly are tumor cells and/or fibroblasts loaded into the system. What's the experimental setting, what cell and tissue culture devices, plates, or slides are used, also for microscopy, etc... none of this is totally clear to me and I doubt it would be to other readers.

Author Response 10: We went in detail on the experimental setting for the 3D µTs production adding a Figure S3 in the Supplementary file. In addition, we better clarify the several steps of 3D µTs in subsection “In vitro 3D Colorectal cancer microtissues fabrication” in the “Material and Methods” section: line 536-582

Figure 1: preparation of the figures just isnt as professional as it should. There are many elements missing in the figure legends, such as complete declaration what are the units shown in 2G, H and I? Cell number/area... in what dimension? collagen content - in which measure precisely? what exactly is CAD? and what is CAD analysis ???

Author Response 11: We thank the Reviewer for his/her kind comment. We added the units in Figure 1G, H, I. Regarding the collagen fraction analysis, the collagen content is expressed by the percentage of Collagen synthesized, calculated as the ratio between bright pixels (collagen portion) to total of black pixel (non-collagen portion) and bright pixel in the selected ROI. Instead, the Collagen Assembly Degree (CAD) measure the intensity of collagen fibers that is proportional to the degree of assembly of the newly synthesized collagen. It is calculated as ratio between the sum of pixel intensities with the corresponding gray value interval from 1 to 255 and total of pixel intensities. In this respect, we described the Collagen fraction analysis and Collagen Assembly Degree in the subsection of the “Materials and Methods” section (line 597-615).

Fig. 2: The exact same issues also apply for the very similar Fig. 2 which lacks the same details.

Author Response 12: We added the units in Figure 2G, H, I.

Fig.3: again, here the same issues with details that should be explained more thoroughly for the reader to be able to understand what exactly is measured

Author Response 13: We added the units in Figure 2G, H, I.

Round 2

Reviewer 2 Report

The manuscript has become easier to read, since it has been rearranged; as recommended by both reviewers. Also the order of the figures has improved the understanding of the narrative of the paper. While the reader really had to struggle a bit to understand the nature of the 3 different µTs compared, this is now more clear. 

There are still a few smaller issues that I would like to see commented on. 

Figure 2 and 3 and 4: The SHG signal is still difficult to see. I can understand why the authors didn't simply stain for human collage in gelatin beads; but could this be tried with e.g. collagen IV? Fibronectin? Anything that comes from the fibroblasts? O(see comments on vimentin and cytokeratin stainings below). 

I also wonder: are the normal fibroblasts penetrating into the gelatin beads and remodeling them? And how fast does it happen (period days 1-4 is suggested). WHat is changing in time course? The second-harmonic generation (SHG) signal just isn't very strong and therefore not utterly convincing. And are the NAFs/CAFs entering the beads, how fast does it happen, do they remodel them from the outside in or even the inside out? Instead, it looks like the action of CAFs results in a clumping/clustering-together of the beads in larger aggregates.

It would be helpful to actually SEE the cells inside the beads and to be able to more clearly distinguish the. I appreciate the GFP-labelen HCT116 cells, but the fibros remain invisible throughout these early images.  Even if somewhat sophisticated methods are used such as CAD (collagen assembly degree ) analysis, a good and simple high resolution picture highlighting different cell types in contact still tells more than 1000 words. 

As previously recommended, it would also help to simply counter-stain cell nuclei with DNA stains in all the figures, as it is done for Fig. 6 and 7, why not for the earlier figures 1-4?  That way, it still remains somewhat unclear if the dark dots inside the gelatin beads are cells (or their nuclei) or if these are structural inhomogeneities of the beads. 

Another simple IF stain would clarify these things immediately, and the authors are indeed using such stains/markers in Fig. 6 and 7. Those are alpha-smooth muscle actin, or FAP alpha for the activated versus normal fibroblasts. Fine; but at the earlier stages, and earlier figures, the reader would be satisfied with more simple, general cell-specific markers including vimentin for mesenchymal cells, versus pan-cytokeratin stain, for the epithelial tumor cells.  And maybe show the changing dynamic composition of the beads at days 4, 8 and 12. 

Instead, the authors REALLY like the SHG signal. I have always considered that as indirectly informative and cell-based markers are superior when it comes to understanding the tissue formation process. Which is what the authors really want to claim. 

The rest of the manuscript, also the discussion, are now rather clear, I cannot see any problems with these parts. The discussion has been essentially rewritten, and is now more clear. 

There are still a relatively large number of grammatical errors which I have not bothererd to point out the last time.  And its not possible to point them out here, without spending an entire day over the manuscript. The authors should actually use the artivcle "the" in conjunction with the abbreviations TME and ECM, hats very often missing throughout the text. Its only one example of grammar changes that are due in article production and will take a while. 

Author Response

The manuscript has become easier to read, since it has been rearranged; as recommended by both reviewers. Also the order of the figures has improved the understanding of the narrative of the paper. While the reader really had to struggle a bit to understand the nature of the 3 different µTs compared, this is now more clear. 

There are still a few smaller issues that I would like to see commented on. 

Figure 2 and 3 and 4: The SHG signal is still difficult to see. I can understand why the authors didn't simply stain for human collage in gelatin beads; but could this be tried with e.g. collagen IV? Fibronectin? Anything that comes from the fibroblasts? O(see comments on vimentin and cytokeratin stainings below). 

We apologized if the Referee does not appreciate the SHG imaging we performed. We used SHG to capture triple helical collagen molecules in the microtissue and to quantify SHG signals for collagen content and distribution in the samples. SHG is widely used to image collagen fibrils. The non-centrosymmetric structure of some collagen fibers give rise to SHG signals and can be used for imaging. For instance, Collagen I gives very strong SHG signals (due to its non-centrosymmetric structure) Collagen IV does not produce SHG. We are interested in detecting Collagen I fibrils and fibers as main component of ECM, so we appreciated the use of this technique. To satisfy the Referee, we tried to improve the SHG signal in the images 2, 3 and 4, but we have to admit that in our opinion it is quite clear the presence of collagen fibers. We acknowledged that SHG does not detect the fibroblasts, but it is an undirect signal of their presence since it is a proof of the biosynthetic activity of fibroblasts, and it is what we want to demonstrate. Indeed, we looked for the best culture conditions allowing the cells to deposit collagen. We observed the presence of collagen fibers in the 3D CRC day4 mTp especially when ascorbic acid is added as indicated by white triangles in the images. In contrast the light grey signal of the microcarrier is noise, indeed since gelatin is composed by denatured collagen, it does not produce SHG. In this regard, we reported below a paragraph extracted by Schmeltz, M.; Robinet, L.; Heu-Thao, S.; Sintès, J. M.; Teulon, C.; Ducourthial, G.; Mahou, P.; Schanne-Klein, M. C.; Latour, G., Noninvasive quantitative assessment of collagen degradation in parchments by polarization-resolved SHG microscopy. Sci Adv 2021, 7, (29).

“The SHG signals collected by nonlinear optical microscopy in biological tissues are specific for dense non-centrosymmetric assemblies of aligned peptide bonds, as in fibrillar collagen, where the a chains are tightly aligned within triple helices, which are, in turn, tightly aligned within fibrils. In contrast, gelatin (denatured collagen) forms a centrosymmetric and low-density network of single helices, connected by small triple-helical domains, and accordingly, it exhibits no SHG signal. During collagen degradation, SHG signal is thus lost because of the alteration of the collagen hierarchical structure.”

I also wonder: are the normal fibroblasts penetrating into the gelatin beads and remodeling them? And how fast does it happen (period days 1-4 is suggested). WHat is changing in time course? The second-harmonic generation (SHG) signal just isn't very strong and therefore not utterly convincing. And are the NAFs/CAFs entering the beads, how fast does it happen, do they remodel them from the outside in or even the inside out? Instead, it looks like the action of CAFs results in a clumping/clustering-together of the beads in larger aggregates.

Normal fibroblasts enter the porous of the microbeads, at first adhere and proliferate (1-4 days) and then synthesize and deposit collagen (from day 4 onward). Indeed, the collagen deposition is evident in the picture Fig. 2C in which the white arrow indicates bright grey signal inside a microbead porous. From day 4 to day 12 it is possible to observe the evolution from single cell seeded microbead to the final microtissue configuration that is a cluster composed by gelatin microbeads, cells and ECM. The aggregation is the results of fibroblasts biosynthetic and proliferation activity, as correctly observed by referee.

We have previously explained [1][2][3] the bioengineered strategy we established to produce connective microtissue, based on a dynamic spinner culture that allows fibroblasts to seed, proliferate and deposit collagen, on the surface and in the porosity of gelatin microbeads. In these conditions, the single cell-seeded microbead can join due to cell-cell and cell-ECM interaction forming microtissue.

We found that cells begin to deposit assembled collagen from day 4 onward. We detected the activation of normal fibroblasts towards CAF phenotype at day 12 in agreement with our previous work [3]. Both normal fibroblasts and CAF remodel their own ECM as demonstrated in the figure 5.

[1] Imparato, G.; Urciuolo, F.; Casale, C.; Netti, P. A., The role of microscaffold properties in controlling the collagen assembly in 3D dermis equivalent using modular tissue engineering. Biomaterials 2013, 34, (32), 7851-7861.

[2] Brancato, V.; Garziano, A.; Gioiella, F.; Urciuolo, F.; Imparato, G.; Panzetta, V.; Fusco, S.; Netti, P. A., 3D is not enough: Building up a cell instructive microenvironment for tumoral stroma microtissues. Acta Biomaterialia 2017, 47, 1-13.

[3] Brancato, V.; Comunanza, V.; Imparato, G.; Corà, D.; Urciuolo, F.; Noghero, A.; Bussolino, F.; Netti, P. A., Bioengineered tumoral microtissues recapitulate desmoplastic reaction of pancreatic cancer. Acta Biomaterialia 2017, 49, 152-166.

It would be helpful to actually SEE the cells inside the beads and to be able to more clearly distinguish the. I appreciate the GFP-labelen HCT116 cells, but the fibros remain invisible throughout these early images.  Even if somewhat sophisticated methods are used such as CAD (collagen assembly degree ) analysis, a good and simple high resolution picture highlighting different cell types in contact still tells more than 1000 words. 

As previously recommended, it would also help to simply counter-stain cell nuclei with DNA stains in all the figures, as it is done for Fig. 6 and 7, why not for the earlier figures 1-4?  That way, it still remains somewhat unclear if the dark dots inside the gelatin beads are cells (or their nuclei) or if these are structural inhomogeneities of the beads. 

Another simple IF stain would clarify these things immediately, and the authors are indeed using such stains/markers in Fig. 6 and 7. Those are alpha-smooth muscle actin, or FAP alpha for the activated versus normal fibroblasts. Fine; but at the earlier stages, and earlier figures, the reader would be satisfied with more simple, general cell-specific markers including vimentin for mesenchymal cells, versus pan-cytokeratin stain, for the epithelial tumor cells.  And maybe show the changing dynamic composition of the beads at days 4, 8 and 12. 

Again, we apologized if in the figure 2-4 the presence of fibroblasts is not highlighted by a specific marker but only, indirectly, by the presence of neo formed collagen, evidently deposited by fibroblasts. However, our aim was to detect the culture conditions that maximize collagen deposition and assembly, for this reason we used SHG imaging and performed quantitative analyses evaluating collagen fraction and collagen assembly degree. The dark dots inside the gelatin beads are the porous of the microbeads, they have a dimension raging between 10 and 20 mm, so they cannot be the cell’s nuclei.

Instead, the authors REALLY like the SHG signal. I have always considered that as indirectly informative and cell-based markers are superior when it comes to understanding the tissue formation process. Which is what the authors really want to claim. 

We admit that we really like SHG imaging that we used in several papers we published to demonstrate the (not trivial) presence of de novo synthesized collagen network in our connective bioengineered tissue. We (but also many other researchers in bioengineering) found that SHG microscopy is a useful tool for studying key facets of collagen remodeling and is an attractive alternative to conventional or fluorescent-based histology for studying tissue composition and visualizing the molecular structure of collagen due to its label-free nature, high sensitivity, and specificity. Once again, we apologized, we did not perform traditional IF staining for detecting fibroblasts markers as vimentin, but our main aim was to detect the fibrillar collagen deposition at the different time points and once established the condition in which the collagen deposition was more favorite investigate fibroblasts activation by IF as reported in the figure 6 and 7 appreciated by Referee. We thank referee for his/her observation, and we treasured his/her suggestion for the next experiments.

The rest of the manuscript, also the discussion, are now rather clear, I cannot see any problems with these parts. The discussion has been essentially rewritten and is now more clear. 

We thank Referee for appreciating our revision

There are still a relatively large number of grammatical errors which I have not bothererd to point out the last time.  And its not possible to point them out here, without spending an entire day over the manuscript. The authors should actually use the artivcle "the" in conjunction with the abbreviations TME and ECM, hats very often missing throughout the text. Its only one example of grammar changes that are due in article production and will take a while. 

We apologized for grammatical errors we checked throughout the manuscript to improve the grammar.